# BiXSE: Improving Dense Retrieval via Probabilistic Graded Relevance Distillation

**Christos Tsirigotis***
Université de Montéal
Mila - Quebec AI Institute

**Vaibhav Adlakha**
McGill University
Mila - Quebec AI Institute

**João Monteiro**
Apple MLR

**Aaron C. Courville**
Université de Montéal
Mila - Quebec AI Institute
IVADO, Canada CIFAR AI Chair

**Perouz Taslakian**
ServiceNow Research

## Abstract

Neural sentence embedding models for dense retrieval typically rely on binary relevance labels, treating query-document pairs as either relevant or irrelevant. However, real-world relevance often exists on a continuum, and recent advances in large language models (LLMs) have made it feasible to scale the generation of fine-grained graded relevance labels. In this work, we propose **BiXSE**, a simple and effective pointwise training method that optimizes binary cross-entropy (BCE) over LLM-generated graded relevance scores. BiXSE interprets these scores as probabilistic targets, enabling granular supervision from a single labeled query-document pair per query. Unlike pairwise or listwise losses that require multiple annotated comparisons per query, BiXSE achieves strong performance with reduced annotation and compute costs by leveraging in-batch negatives. Extensive experiments across sentence embedding (MMTEB) and retrieval benchmarks (BEIR, TREC-DL) show that BiXSE consistently outperforms softmax-based contrastive learning (InfoNCE), and matches or exceeds strong pairwise ranking baselines when trained on LLM-supervised data. BiXSE offers a robust, scalable alternative for training dense retrieval models as graded relevance supervision becomes increasingly accessible.

## 1 Introduction

Training sentence embedding models for dense retrieval, as in DPR (Karpukhin et al., 2020) or Sentence-BERT (Reimers & Gurevych, 2019), typically relies on contrastive learning (Oord et al., 2018, InfoNCE) with binary relevance labels – where query-document pairs are marked as either relevant (positive) or irrelevant (negative). Models are trained to produce similar embeddings for query-positive pairs and dissimilar embeddings for query-negative pairs. While effective, this binary approach can be limiting, as real-world relevance often exists on a continuum rather than as a strict yes/no. For instance, a document might be partially relevant to a query, yet traditional contrastive training would treat it as equally irrelevant as a completely unrelated document. This issue is particularly pronounced when using mined "hard negatives", documents typically selected among the top-ranking retrieval candidates according to an initial retriever model and labeled as irrelevant (Karpukhin et al., 2020; Wang et al., 2022b; Lee et al., 2024b; de Souza P. Moreira et al., 2024). While hard negatives are crucial for effective learning (Lee et al., 2024b), they are uniformly labeled as having zero relevance, even if some may actually be partially relevant, introducing false negatives into the training data (Ni et al., 2021; Qu et al., 2021). Thus, binary simplification provides incomplete supervision – with moderately relevant passages receiving no credit

---

*Correspondance to: Christos Tsirigotis <tsirigoc@mila.quebec >, Perouz Taslakian <perouz.taslakian@servicenow.com >. Code release at: https://github.com/tsirif/BiXSE.

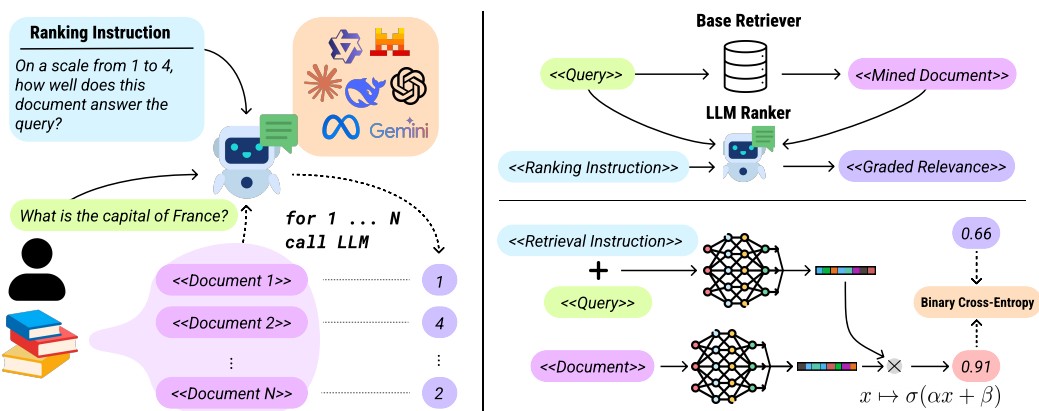

Figure 1: BIXSE amortizes the expensive inference process of an effective graded LLM ranker into the training of a dense retrieval model via a simple binary cross-entropy loss.

– and produces noisy datasets containing false negatives, potentially over-penalizing the model during training.

These limitations can be addressed by using graded relevance scores that quantify the degree of relevance between text pairs. This can be achieved for instance via ordinal scales which capture varying relevance levels (e.g. integers 0–3 or 1–5). Historically, graded relevance evaluation has been central in the information retrieval community for assessing system quality (Järvelin & Kekäläinen, 2000; 2002; Sakai, 2021). For example, in the Deep Learning track of the *Text REtrieval Conference (TREC-DL) 2023* (Craswell et al., 2024), systems were evaluated based on how closely their scores align with human judgments across graded relevance levels: 0 (*'Irrelevant'*), 1 (*'Relevant topic, but does not contain the answer'*), 2 (*'Highly relevant, partial or unclear answer'*), and 3 (*'Perfectly relevant, exact answer'*). On the other hand, manually annotating text pairs with graded relevance is labor-intensive, restricting dataset scalability beyond evaluation benchmark sizes.

Early efforts to provide with scalable graded relevance signals involved training cross-encoding rankers on annotated data, followed by using their judgments as pseudo-labels to be distilled into bi-encoders (Hofstätter et al., 2021; Santhanam et al., 2022; Chen et al., 2024; Huang & Chen, 2024). More recently, prompting LLMs like GPT-4 to serve as zero-shot rankers has yielded surprisingly strong results, sometimes surpassing supervised retrievers (Sun et al., 2023). These LLM-based rankers can produce nuanced relevance judgments at scale, unconstrained by binary decisions. For instance, an LLM can be prompted with: "On a scale from 1 to 5, how well does this passage answer the query?" or offered options like '*Not relevant*', '*Partially relevant*', and '*Highly relevant*'. Zhuang et al. (2024) demonstrated that permitting GPT-4 or PaLM to select among fine-grained relevance labels (instead of binary yes/no decisions) significantly enhances ranking accuracy by more effectively handling borderline cases and reducing labeling noise.

Prior to the emergence of LLM-based relevance scoring, graded supervision was primarily leveraged via training with listwise or pairwise objectives (Burges et al., 2005; Wang et al., 2018; Qu et al., 2021; Reddi et al., 2021; Huang & Chen, 2024). However, such methods often rely on labeling multiple hard negative documents per query, significantly increasing annotation cost when using powerful LLMs as teachers, which hinders scalability. The opportunity of LLM-generated large-scale graded relevance data, however, calls for revisiting training objectives to align with the increased costs of quality graded relevance data. In this work, we propose **Binary Cross-Entropy Sentence Embeddings** (BIXSE), a simple pointwise training method that directly optimizes a binary cross-entropy (BCE) loss on graded relevance scores. BIXSE interprets graded relevance scores as probabilities within the range $[0, 1]$ to represent relevance continuity from completely irrelevant (0) to absolutely relevant (1). Unlike pairwise or listwise objectives, which rely on multiple supervised comparisons per query, BIXSE scales efficiently by enabling competitive performance by just

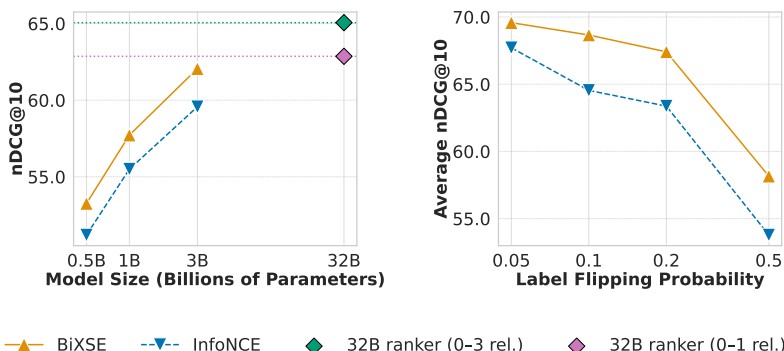

Figure 2: **Left:** BIXSE vs standard InfoNCE training of QWEN2.5 dense encoders. Our method outperforms the standard softmax-based contrastive recipe (InfoNCE) across model sizes when training on the LightBlue dataset, which contains multilingual query-document pairs graded by a zero-shot QWEN2.5-32B-INSTRUCT ranker. We measure nDCG@10 on a 100k random samples of TREC-DL qrels collected from 2019 to 2023. **Right:** BIXSE displays improved robustness to noise compared to InfoNCE models. We train MODERNBERT models on the open portion of E5 dataset and control the chance of flipping the binary label between the positive and the hard negative document. We report the average nDCG@10 score on a subsampled version of BEIR.

using a single labeled query-document pair per query, while capturing structure implicitly via in-batch negatives.

We validate BIXSE through extensive experiments across retrieval and sentence embedding benchmarks, demonstrating consistent gains over standard InfoNCE objectives. Furthermore, BIXSE is, to the best of our knowledge, the first pointwise training method to consistently match or outperform strong pairwise ranking baseline losses when training on LLM-labeled graded relevance datasets. BIXSE scales across architectures and languages, approaches the performance of much larger zero-shot LLM-based rankers, and exhibits improved robustness to label noise compared to InfoNCE. Notably, it benefits from learning across a wider spectrum of graded relevance and achieves peak performance even without aggressive data filtering, making it a strong and efficient alternative for training dense models on LLM-generated supervision. As graded relevance becomes increasingly easy to generate, we argue that BIXSE offers a practical, robust, and scalable training paradigm for the next generation of dense retrieval systems.

## 2 Related Work

**Label Noise in Retrieval Datasets and Mining Hard Examples.** Retrieval datasets generally consist of a query and a corresponding document with a relevance score. In the simplest case, the dataset only consists of positive documents (relevance score of 1.0). However, the dataset curation can introduce label noise in the dataset (Qu et al., 2021; Wang et al., 2022). For example, to select positive passage using question-answering dataset, Karpukhin et al. (2020) declare the highest-ranked passage from BM25 that contains the answer string as the positive passage. This process generates false positives as it is possible to match passages that are not relevant but include the answer string. Label noise also exists in form of false negatives. For example, Qu et al. (2021) report that within the top-retrieved passages for MSMARCO (Bajaj et al., 2018) dataset, 70% of them are actually positives while not being explicitly marked as positive. Reasonably, label noise is further amplified when retrieved passages are used as hard negatives during training (Xiong et al., 2021; de Souza P. Moreira et al., 2024). We thus argue that utilizing training objectives that are more robust to noise can lead to downstream improvements in text encoders.

**Training Dual Encoders with Synthetic Data.** Many recent approaches have turned to synthetic data to get high-quality and diverse training samples at scale (Zhang et al., 2023a;

Wang et al., 2023; Dai et al., 2023; Muennighoff et al., 2024). E5 (Wang et al., 2023) proposed a two-step generation, where LLM first brainstorms potential downstream retrieval tasks, and then generates samples for each of those tasks. Promptagator (Dai et al., 2023) prompts an LLM to generate synthetic queries for existing passages and trains dense retrievers on these generated pairs. More recent approaches like Gecko (Lee et al., 2024b) and Gemini Embeddings (Lee et al., 2025) use LLMs for both sample generation and filtering, yielding high quality training dataset. The strong results of Gemini on MMTEB demonstrate the effectivess of LLM based dataset generation and filtering. Despite this progress, synthetic data generation has predominantly treated relevance as a binary signal, as documents are either relevant or not. The use of graded relevance scores, such as those common in TREC and traditional IR evaluation, remains largely unexplored as a training signal. Our work investigates this underutilized axis of synthetic supervision in training dense retrieval models.

**Distillation Methods for Dense Retrieval and Ranking Models.**    In addition to data filtering, another key way to improve dual encoders is via distillation from stronger re-rankers. The idea is to train the dense retriever to imitate the outputs of a more powerful but slower model (the teacher). Typically, the teacher is a cross-encoder that can deeply inspect each query–document pair, producing superior relevance judgments.

Several loss formulations have been proposed to train with relevance judgements from the teacher model. Cheng et al. (2023) introduce "soft" InfoNCE, where regular one-hot label in InfoNCE is replaced by the soft labels from the teacher. We showcase in Section 5 that BIXSE outperforms this loss formulation. MarginMSE (Hofstätter et al., 2021) minimizes the mean squared error between teacher and student score margins for positive–negative pairs. A limitation of this approach is its inability to profit from in-batch negatives, as performance tends to decay with increased number of negative documents per positive ones. BIXSE fixes this by incorporating a special logit bias term targetted at counteracting this effect.

Pairwise training objectives have also been widely used for dense retrieval. RankNet (Burges et al., 2005), and its recent adaptations such as PairDistill (Huang & Chen, 2024), supervise the model by comparing pairs of documents for the same query. Instead of assigning an absolute relevance score to each document independently, these methods teach the model to prefer one document over another based on their relative graded relevance. In LambdaLoss (Wang et al., 2018), individual pairwise losses are further weighted according to their estimated impact on evaluation metrics such as nDCG. While effective, these pairwise methods tend to achieve their best performance when each query is compared against multiple other labeled documents. This increases annotation costs when labels come from expensive cross-encoders or LLM rankers. Because BiXSE applies binary cross-entropy loss at the pointwise level, it scales more naturally to large datasets and varied supervision sources. It enables training with fewer graded annotations per query, making it well-suited to scenarios where labeling costs are a concern. Our experiments show that BiXSE is competitive against strong pairwise baselines, while offering a lower annotation cost per query.

RocketQA(Qu et al., 2021) and its successor RocketQAv2 (Ren et al., 2021) employ an iterative listwise training procedure: a cross-encoder teacher is used to label a large set of positives and hard negatives, and the dual encoder is jointly trained on those via minimizing a KL on the document batch likelihoods. Similarly, RankDistill (Reddi et al., 2021) encourages the student to reproduce the teacher's top-$k$ rankings, by using the teachers' scores to construct targets for the document batch likelihoods. However, these listwise or teacher-in-the-loop approaches tend to be computationally expensive and less scalable – running a cross-encoder over many candidate passages for every query (often repeatedly in training) incurs a large cost. Moreover, while listwise objectives seem desirable as they align with ranking metrics like nDCG, they assume access to complete and consistent slates of query-document pairs – an assumption that could be violated in practice due to noisy, sparse, or inconsistent supervision from LLMs or heuristics. In contrast, BiXSE's pointwise formulation offers a scalable and robust alternative, enabling effective learning from graded signals without requiring coherent global rankings.

## 3 Preliminaries

We present the notation needed to expose our training procedure by contrasting it to the typical multi-class contrastive learning of dense encoders, or InfoNCE (Oord et al., 2018) as it is commonly referred to in the context of self-supervised learning literature. Let $f$ be the text encoder we would like to train, which, in general, is a neural network that takes text as input and outputs a fixed-dimensional vector. During training, the encoder is presented with batches of $B$ text tuples $\mathcal{B} := \{(q_i, d_i^+, d_i^-)\}_{i=1}^{B}$. Let $q_i$ be a query associated with a positive document $d_i^+$ and potentially a (hard) negative document $d_i^-$. We define $\boldsymbol{x}$ (boldface math font) to be the $L^2$-normalized embedding pooled from the text encoder $f$, that is $\boldsymbol{x} := \frac{f(x)}{\|f(x)\|}$. To train with the InfoNCE objective, we first separately compute the normalized embeddings for each query $\boldsymbol{q}_i = \frac{f(q_i)}{\|f(q_i)\|}$, and similarly for each positive document $\boldsymbol{d}_i^+$, and each hard negative document $\boldsymbol{d}_i^-$. Now, consider a scoring function $s : (q, y) \mapsto \mathbb{R}$ between a pair $(q, d)$ of text according to the dense encoder $f$. Typically, this is taken as a scaled dot-product between the normalized embeddings, $s(q, d) := \alpha \, \boldsymbol{q}^\top \boldsymbol{d}$, for scalar hyperparameter $\alpha > 0$ that acts as an inverse temperature. Then, the training loss for $f$ can be expressed in terms of the function $s(x, y)$ as

$$\mathcal{L}_{\text{InfoNCE}} = \frac{1}{B} \sum_{i \in [B]} -\log \frac{\exp(s(q_i, d_i^+))}{\sum_{j \in [B]} \exp(s(q_i, d_j^+)) + \exp(s(q_i, d_j^-))} . \tag{1}$$

This loss function corresponds to the negative log-likelihood of classifying each query $q_i$ into one of $2B$ possible classes, where each class represents a document from $\mathcal{B}$. The target class for query $q_i$ is always determined by the positive $d_i^+$ for every $i \in [B]$. The alternative classes are defined by the hard negative document $d_i^-$ and *in-batch* negatives[1], meaning that positives $d_j^+$ and hard negatives $d_j^-$ associated by the dataset with other queries $q_j, j \neq i$ are also considered as negative documents with respect to $q_i$. Karpukhin et al. (2020) show that this assumption is essential for achieving strong performance on downstream tasks, particularly in the absence of hard negative documents.

InfoNCE contrastive learning has been the de-facto paradigm for training dense retrievers and generalist sentence embedders. The training loss formulation in Equation (1) relies on *binary relevance*, i.e. a boolean assignment (relevant/positive or non-relevant/negative) for every query-document pair.

## 4 BiXSE: Binary Cross-Entropy Sentence Embeddings

The search for a more fine-grained learning signal starts within the information retrieval community (Järvelin & Kekäläinen, 2000; 2002; Sakai, 2021) and its efforts to define a query-document relevance that is more closely aligned with human judgement. Such reliance on human annotations has hindered the scalability of graded relevance labeled datasets. However, recent LLMs have demonstrated strong zero-shot ranking capabilities (Sun et al., 2023; Zhuang et al., 2024), surpassing dense retrieval models based on human judgement, without requiring additional fine-tuning on ranking data. Although automated, querying LLMs for generating a quality dataset remains an expensive procedure constrained by financial considerations. Therefore, it is crucial to develop scalable solutions that still provide competitive advantages when training with graded relevance data. As discussed in Section 2, listwise approaches (Qu et al., 2021; Ren et al., 2021; Reddi et al., 2021) require labeling multiple documents per query by the LLM, which constraints the scalability of data along another qualitative axis – such as the number of queries contained in a dataset – given a fixed LLM token consumption budget. At the same time, many of those query-document

---

[1]or *cross-batch* if they belong to mini batches sampled on different computational devices of a data parallel training trial. In the paper, we use *in-batch* and *cross-batch* interchangeably to mean that all other unlabeled query-document combinations are used as negatives.

pairs are filtered out in state-of-the-art pipelines (Lee et al., 2024b; 2025) based on LLM-generated graded relevance, leading to token waste. By exploring a pointwise solution, our goal is to develop a scalable, token-efficient and effective method for leveraging graded relevance information.

Our work focuses on improving the effectiveness of models trained on a dataset with graded relevance scores. In contrast to the setting in Section 3, batches from this dataset may now consist of a query associated with a set of $K > 0$ documents, and a score that indicates the relevance of each document. More formally, for a given query $q_i$ let $d_i^{(k)}$ be the $k^{th}$ document associated with $q_i$, for each $i = 1, 2, \ldots, B$ in the batch. We let $z_i^{(k)} \in [0, 1]$ denote the continuous relevance score between query $q_i$ and document $d_i^{(k)}$, with 0 being most relevant and 1 absolutely relevant. Then, a batch $\mathcal{B}$ contains a set of tuples $(q_i, \{(d_i^{(k)}, z_i^{(k)})\}_{k=1}^K)$. This setup generalizes the one described in Section 3 where relevance is a binary concept: previously, a document is either relevant to a query (denoted as $d_i^+$) or is not ($d_i^-$), with no notion of *degree* of relevance. Our proposed setup, in which relevance degree is represented as the score $z_i$, is a generalization of the binary setup with $z_i$ set to 1 for positive documents $d_i^+$ and to 0 for hard negative documents $d_i^-$.

Given that we are studying a pointwise solution, we restrict our discussion to the setup where each query $q_i$ is relevant to single document ($K = 1$). This does not limit generality, as our pointwise loss can be applied to additional query-document-relevance triplets if needed. Accordingly, we thus redefine a batch as $\mathcal{B} := \{(q_i, d_i, z_i)\}_{i=1}^B$ as a set of query-document-relevance triplets during training.

One way we can practically acquire graded relevance labels at scale is by asking an LLM to output an ordinal judgement about a query-document pair given a relevance definition in the instruction (Zhuang et al., 2024), as visualized in Figure 1. Having access to the logits of the model furthermore enables us to convert a finite number of discrete ordinal judgements to continuous ones by using the constrained predicted probability among the discrete score options to average the possible ordinal outcomes. More formally, given a relevance definition that contains a set of possible discrete scores $S = \{0, 1, \ldots, N\}$, we can convert these scores to continuous ones $z := \sum_{s \in S} s \, p_{\text{LLM}}(s|q, d)$ (Lightblue, 2025). Then, $z$ can be converted to $[0, 1]$ via a (rank-preserving) increasing function, such as an affine transformation $\frac{z}{N}$.

As before, batches of tuples in $\mathcal{B}$ are presented to the encoder during training. Similarly as before, each query and document are separately processed via encoder $f$ to produce $L^2$-normalized embeddings $\boldsymbol{q}_i$ and $\boldsymbol{d}_i$ respectively. This time we opt for a different choice of scoring function, which is defined as $s(q, d) := \alpha \, \boldsymbol{q}^\top \boldsymbol{d} + \beta$ for some logit scale and bias parameters $\alpha > 0$ and $\beta$. We enhance the loss formula from Equation (1) by incorporating a crucial logit bias term to improve performance. We make use of in-batch negatives by supposing labels $z_{i,j} = 0$ for $i \neq j$, otherwise we use the graded relevance score from the batch $z_{i,i} = z_i$. Let $\sigma(x) := (1 + \exp(-x))^{-1}$ be the sigmoid/logistic function. Then, the encoder $f$ is trained to minimize the binary cross-entropy (BCE) loss

$$L_{\text{BiXSE}} = -\frac{1}{B} \sum_{i \in [B]} \sum_{j \in [B]} z_{i,j} \log \sigma(s(q_i, d_j)) + (1 - z_{i,j}) \log \sigma(-s(q_i, d_j)). \qquad (2)$$

The behavior of this loss function depends on the value of $z_{i,j}$. When $z_{i,j} > 0.5$, we want the sigmoid to output a value closer to 1, implying that the normalized embeddings $\boldsymbol{q}_i$ and $\boldsymbol{d}_j$ need to be more aligned.

**The role of logit bias $\beta$.** The use of in-batch negatives introduces a strong label imbalance: for each query $q_i$, the model sees one (likely) positive document $d_i$ and $B - 1$ negatives, skewing the label distribution toward zero. This imbalance becomes more pronounced with larger batch sizes, making it easier for the encoder $f$ to minimize the loss by predicting low relevance scores across the board. We interpret the logit bias $\beta$ in Equation (2) as a mechanism to correct for this skew by modeling the marginal label distribution. Since $\beta$ is not conditioned on query-document pairs, it cannot learn relevance itself, but it can only

*Models finetuned on English-only data*

| Categories | BEIR | MTEB(eng, v2) Retr. | TREC-DL 19-23[†] | MTEB(eng, v2) All |
|---|---|---|---|---|
| MODERNBERT-BASE | | | | |
| INFONCE | 41.32 | 39.17 | 42.32 | 51.79 |
| BIXSE | 42.29 (+2.3%) | 41.11 (+5.0%) | 47.67 (+12.6%) | 55.66 (+7.5%) |
| META-LLAMA-3.2-1B-INSTRUCT | | | | |
| INFONCE | 49.31 | 48.76 | 58.39 | 62.87 |
| BIXSE | 49.51 (+0.4%) | 50.60 (+3.8%) | 59.85 (+2.5%) | 63.13 (+0.4%) |

*Models finetuned on multilingual data*

| Categories | BEIR | MTEB(multi, v1) Retr. | TREC-DL 19-23[†] | MTEB(multi, v1) All |
|---|---|---|---|---|
| QWEN2.5-0.5B-INSTRUCT | | | | |
| INFONCE | 38.25 | 50.24 | 55.29 | 48.98 |
| BIXSE | 73.91 (+6.0%) | 51.90 (+3.3%) | 55.71 (+0.8%) | 49.68 (+1.4%) |
| QWEN2.5-1.5B-INSTRUCT | | | | |
| INFONCE | 43.66 | 55.00 | 57.93 | 52.64 |
| BIXSE | 48.08 (+10.1%) | 58.45 (+6.3%) | 59.95 (+3.5%) | 53.59 (+1.8%) |
| QWEN2.5-3B-INSTRUCT | | | | |
| INFONCE | 48.83 | 59.22 | 62.40 | 55.74 |
| BIXSE | 50.55 (+3.5%) | 61.67 (+4.1%) | 64.70 (+3.7%) | 57.02 (+2.3%) |

Table 1: Average aggregated NDCG@10 performance on various retrieval benchmarks of models trained with the same data resources. BIXSE consistently outperforms models trained with the standard softmax-based contrastive learning loss. [†]We benchmark against a corpus formed by collecting all annotated documents in the available qrels from 2019 until 2023.

model the marginal distribution of labels. To encourage this separation of roles, we optimize $\beta$ with a significantly higher learning rate than $f$, ensuring that the encoder $f$ must rely on the actual query-document content to minimize the loss. We study the effect of this design choice in Figure 5 and Appendix 6.

## 5 Experiments

**Base Models.**  For our experiments we finetune a variety of base models seeking to validate the benefits of our approach for different pretraining methods and model sizes. For this reason, we experiment with MODERNBERT (Warner et al., 2024), a bidirectional masked language model (MLM), as well as with META-LLAMA-3.2 (Meta AI, 2024) and QWEN2.5 (Qwen Team, 2024) models exemplifying decoder-as-encoder approaches. As a sentence embedding for the MODERNBERTarchitecture, we use the representation extracted by the standard pooler on top of the beginning-of-sentence token. On the other hand, we convert decoder LLM architecture following the LLM2Vec recipe (BehnamGhader et al., 2024). In particular, we disable the autoregressive masking, thus enabling bidirectional connections among tokens, and we pool token representations into a fixed dimensional sentence embedding by averaging the last layer representations over the query tokens, or the document tokens accordingly. We skip the first MLM finetuning step of LLM2Vec, instead we directly finetune the adapted models with contrastive objectives. Finally, we enable our models to be prompted with task-specific instructions (Su et al., 2023) when we encode queries, allowing for extra flexibility in the representation space.

**Datasets.**  Our experiments span across multilingual and English-exclusive data. We also experiment with data labeled with graded relevance scores, binary and mixed cases, hoping to demonstrate the effectiveness of BIXSE in a backward-compatible manner. For binary relevance data, we use the public portion of the E5 dataset (Wang et al., 2022) reconstructed by Springer et al. (2024). In Table 2 of the Appendix, we describe the composition of training sets and their corresponding tasks. In Section 6 of the Appendix, we provide the list of instructions we used to augment queries. We use the same task instructions as previous works Wang et al. (2023); Springer et al. (2024); BehnamGhader et al. (2024). As for graded

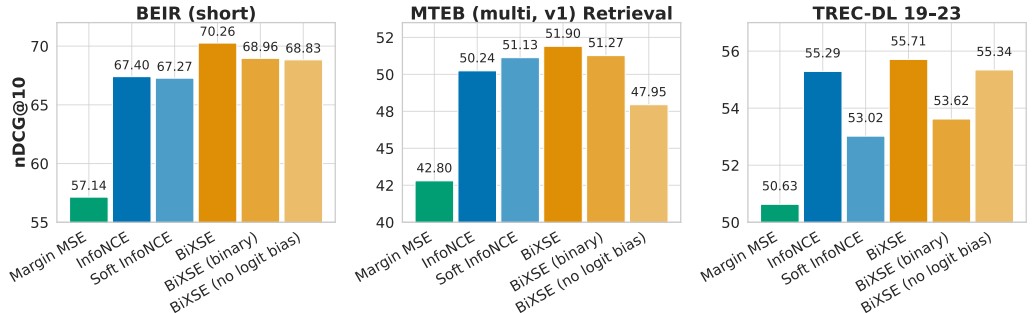

Figure 3: Ablating loss function variants by training QWEN2.5-0.5B-INSTRUCT models; binary cross-entropy loss performs the best when both graded relevance scores and a logit bias in the scoring function are used.

relevance data, we base our experiments on a public dataset (Lightblue, 2025) that we will call *LightBlue* (or LB in abbreviation). LightBlue is a collection of 35 high quality question-answering datasets covering more than 95 languages, whose authors curate for training distilled cross-encoders. For query-document pairs in the collection, the authors ask a QWEN2.5-32B-INSTRUCT-GPTQ-INT4[2] model to zero-shot rank their relevance with a discrete number scaling from 1 to 5. The provide with model logits over the token responses, which we then convert to a continuous label in $[0, 1]$, as we discussed in Section 4. More details about LightBlue dataset are discussed in Appendix 6.

**Benchmarks.** We evaluate our approach on several benchmarks spanning heterogeneous retrieval, graded relevance retrieval, and universal text embedders. BEIR (Thakur et al., 2021) is a zero-shot heterogeneous retrieval evaluation framework encompassing diverse tasks and domains. It consists of 18 different datasets from nine different tasks – Fact checking, citation prediction, duplicate question retrieval, argument retrieval, news retrieval, question answering, tweet retrieval, bio-medical IR, and entity retrieval.

To evaluate our method beyond retrieval, we also consider MMTEB (Enevoldsen et al., 2025) (multilingual version of MTEB Muennighoff et al. (2023)) which covers multiple embedding-based tasks across various languages. MMTEB introduced several benchmarks – MTEB(eng, v2) is the faster zero-shot version of widely popular MTEB benchmark (Muennighoff et al., 2023) which contains 41 English embedding tasks across seven task categories – retrieval, reranking, clustering, pair classification, classification, sentence similarity, and summarization. MTEB(multi, v1) on the other hand consists of 131 tasks from several task categories and languages. We show results on both retrieval and the complete set of MMTEB benchmarks.

Last, we perform evaluations against graded relevance scores reflecting human judgement about query-document relations using publically available data from TREC-DL (Craswell et al., 2024) competitions. We assemble the qrels from all competitions from 2019 till 2023 into a collection of 422612 graded relevance scores over 1988 queries on the MSMARCO (Bajaj et al., 2018) passage corpus.

**Main Results.** We train models on English data consisting of a mixture of binary relevance E5 and graded relevance LB datasets, and multilingual models purely on graded relevance scored data from LB. Table 1 reports NDCG@10 performance of models trained with BIXSE versus InfoNCE across English and multilingual settings, covering both retrieval and sentence embedding tasks. BIXSE consistently outperforms InfoNCE across all base models and benchmarks, with the most substantial gains observed on TREC-DL 2019–2023 highlighting BIXSE's strength in modeling nuanced relevance signals; for instance, BIXSE achieves +12.6% improvement on MODERNBERT-BASE, +3.7% on QWEN2.5-3B-INSTRUCT, and +3.5% on QWEN2.5-1.5B-INSTRUCT. On binary relevance benchmarks such as BEIR and MTEB (Retrieval), BIXSE maintains consistent improvements – e.g., +10.1% on BEIR

---

[2]https://huggingface.co/Qwen/Qwen2.5-32B-Instruct-GPTQ-Int4

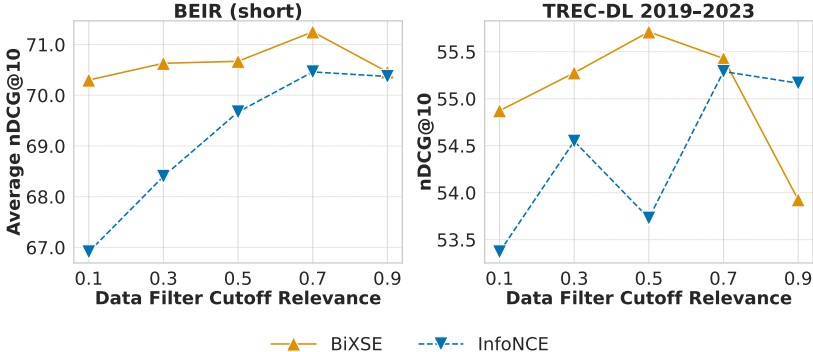

Figure 4: Performance of QWEN2.5-0.5B-INSTRUCT models trained using BIXSE or InfoNCE models under different graded relevance cutoff values for positives. InfoNCE improves as we discard data pairs of lower graded relevance, while BIXSE follows a reverse U-curve trend. Overall the best BIXSE model outperforms the best InfoNCE one, showing that BIXSE effectively enables learning from graded relevance labels, and consequently less token consumption waste during dataset creation.

with QWEN2.5-1.5B-INSTRUCT. Gains also extend to the average of all tasks in MTEB, a general sentence embedding suite, where BIXSE yields up to +7.5% improvement, indicating that training with graded supervision enhances embedding quality beyond retrieval. Altogether, across six model backbones and four benchmark families, BIXSE delivers reliable, architecture-agnostic improvements in both binary and graded settings, validating its core motivation of leveraging LLM-derived graded relevance labels through a simple yet effective binary cross-entropy objective. In Appendix 6, we provide with analytical reports of model performance per individual tasks contained in BEIR or MTEB, whereas in Table 9 of Appendix 6 we provide with analytical model performance results on the top-100 document reranking tasks of TREC-DL for each year separately.

**Comparison to other training objectives.** We further compare BIXSE against alternative training objectives such as Soft InfoNCE (Cheng et al., 2023) and Margin MSE (Hofstätter et al., 2021), as well as ablated variants of BIXSE. While Soft InfoNCE offers a smoother alternative to one-hot probability targets and Margin MSE explicitly models pairwise differences, both underperform BIXSE across all benchmarks, highlighting the advantage of directly optimizing for graded relevance via BCE. Notably, ablations that remove the logit bias or discretize the targets into binary values consistently degrade performance, confirming the importance of BIXSE's full formulation. Finally, we have conducted a comprehensive comparison between BiXSE and other strong pairwise ranking objective baselines, which we detail in Appendix 6. Across both LightBlue and BGE-M3 (Chen et al., 2024) training datasets, BIXSE consistently matches or outperforms alternatives like Pairwise BCE (Burges et al., 2005; Huang & Chen, 2024) and LambdaLoss (Wang et al., 2018), while requiring fewer labeled negatives and enabling more efficient in-batch training. These findings validate the scalability and supervision efficiency of BIXSE, when compared against pairwise alternatives.

**Effective distillation from reranker.** To contextualize BIXSE's performance, in the left plot of Figure 2, we compare our dense encoders to two 32B LLM-based rerankers. We instruct Qwen2.5-32B-Instruct to score query-document pairs from TREC-DL (2019–2023), using prompts aligned with the original human annotation guidelines. In the 0–3 relevance setting, the model uses a four-point relevance scale; in the 0–1 setting, it operates under a simplified binary prompt distinguishing "Irrelevant" from "Relevant, Exact Answer." Despite the model size gap, our 3B BIXSE encoder achieves an nDCG@10 of 62.06 – within 3.0 points of the 32B ranker with full 0–3 prompting (65.06), and just 0.8 points behind the 32B ranker trained with binary prompts (62.86). This result demonstrates that BIXSE can effectively distill LLM supervision into fast, efficient dense retrieval models without sacrificing ranking quality.

**Robustness to noise.** A key motivation for BIXSE is its robustness to labeling noise, which is prevalent in large-scale retrieval datasets due to imperfect negative mining, synthetic supervision, or human annotation inconsistencies. To test this, we simulate controlled label noise by flipping the binary labels between the positive and hard negative documents in the E5 dataset according to a prescribed probability, prior to training. We observe in the right plot of Figure 2 that, as the noise level increases, BIXSE exhibits a markedly more gradual degradation in nDCG@10 compared to InfoNCE. While both methods see performance drops, the curve for BIXSE remains more concave and stable, indicating greater tolerance to noise. We invite the reader to Appendix 6, where we discuss our intuition behind these results.

**Learn best by filtering less.** We investigate how BIXSE and InfoNCE respond to different thresholds for filtering training data by graded relevance. In large-scale datasets built from LLM-generated scores, a common preprocessing step is to discard query-document pairs with low relevance to improve training quality. We vary the minimum relevance cutoff required to retain a pair, and treat the remaining data differently: for InfoNCE, retained pairs are binarized and used as positives; for BIXSE, we preserve their graded scores and train using the full BCE formulation. To control for dataset size, we fix the number of training examples across all cutoff values by subsampling each filtered set to match the smallest resulting dataset (corresponding to the highest cutoff of 0.9). The results, shown in Figure 4, reveal that InfoNCE performance improves monotonically with stricter filtering, suggesting that it benefits from high-confidence positives and sharper separation from negatives. In contrast, BIXSE follows a reverse U-shaped trend, achieving peak performance at a moderate cutoff (e.g., 0.7), indicating that it can effectively learn from a broader range of relevance signals. Crucially, the best BIXSE model outperforms the best InfoNCE model, suggesting that BIXSE not only enables better generalization from nuanced supervision, but also reduces the need for aggressive data pruning—allowing more effective use of available training pairs with less token waste.

## 6 Conclusion

We present BIXSE, a simple and effective training objective that enables dense retrieval models to learn directly from graded relevance labels. By replacing contrastive formulations with a binary cross-entropy loss over continuous targets, BIXSE captures fine-grained supervision more faithfully, yielding consistent performance gains across diverse retrieval and embedding benchmarks. Our results show that BIXSE not only outperforms standard objectives like InfoNCE and Margin MSE, but also narrows the gap to large zero-shot LLM rankers, improves robustness to label noise, and learns better without aggressive filtering. At the same time, it performs competitively with state-of-the-art pairwise losses, while requiring fewer annotated documents per query. As LLMs make graded relevance supervision increasingly accessible, BIXSE provides a practical and scalable solution for distilling this supervision into fast and generalizable dense encoders.

## Acknowledgments

This work was done while Christos Tsirigotis and Vaibhav Adlakha were interning at ServiceNow Research, Montréal. For João Monteiro, work was done prior to joining Apple MLR. Christos acknowledges funding from the Mitacs Accelerate program, as well as from Aaron Courville's CCAI chair and CRC chair.

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

# Appendix

**Extended Related Work**

**Neural Dense Retrieval.** Classical retrieval methods like TF-IDF and BM25 (Robertson & Zaragoza, 2009) represent texts as sparse, high-dimensional vectors based on lexical overlap, effectively capturing keyword matches but lacking deeper semantic understanding. Neural dense retrieval methods address this by embedding texts into dense semantic vector spaces using pre-trained language models (Karpukhin et al., 2020; Xiong et al., 2021).

Dense retrieval methods broadly fall into two categories based on their trade-offs between computational efficiency and retrieval accuracy: bi-encoders (dual encoders) and cross-encoders. Bi-encoders encode queries and documents independently, enabling efficient approximate nearest neighbor (ANN) searches suitable for large-scale retrieval (Ni et al., 2022; Li et al., 2023; Wang et al., 2022; BehnamGhader et al., 2024; Lee et al., 2024a), but they fail to capture fine-grained contextual understanding. Cross-encoders jointly encode query-document pairs, providing more accurate relevance estimates but at higher computational costs, limiting their scalability. The simplest example of cross-encoder is feeding query-document pairs into large language models (LLMs) to explicitly assess relevance (Sun et al., 2023).

With advancements in sentence representation techniques, the research community has expanded its focus from narrow retrieval tasks to general-purpose text embeddings. BEIR (Thakur et al., 2021) is a zero-shot evaluation framework encompassing diverse retrieval tasks and domains whereas MMTEB (Enevoldsen et al., 2025) covers multiple embedding-based tasks like retrieval, classification, and clustering across various languages. To encourage generalization across tasks, it is now common to prepend each input with a natural language task description, guiding the encoder to produce task-specific representations (Su et al., 2023). In this paper, we adopt a similar formulation by conditioning on a prompt that describes the relevance task.

**Representation Learning with Binary Cross-Entropy.** To our knowledge, binary cross-entropy loss in self-supervised representation learning was first explored by Hjelm et al. (2019). Their work introduces Deep InfoMax (DIM), a framework for representation learning that maximizes mutual information between different views of the same image. In its Jensen-Shannon mutual information formulation (Hjelm et al., 2019, See Equation 4, Section 3.1), the DIM loss functions as a binary cross-entropy loss and the logits are computed from pairs of views: pairs derived from the same image are labeled as positive, while pairs from different images are labeled as negative.

More recently, Zhai et al. (2023) introduced SigLIP, a framework that pre-trains vision-language encoders by aligning image embeddings with their corresponding caption embeddings. Compared to its softmax-based anchor classification alternative (Radford et al., 2021, CLIP), pretraining with binary cross-entropy loss in SigLIP has demonstrated better scaling of downstream task performance as the number of in-batch negatives increases. In our work, we adapt the SigLIP objective to fine-tune large language models (LLMs) (BehnamGhader et al., 2024) for general-purpose text embedding tasks.

**Lightblue Reranker Distillation Dataset**

Lightblue (2025) constructed this dataset through a four-step process aimed at creating a diverse, high-quality resource for evaluating query-text relevance. First, they collected queries and associated text passages from 35 publicly available datasets spanning over 95 languages. For datasets lacking hard negatives, they mined them using the BAAI/BGE-M3 embedding model to ensure a challenging contrastive setup. Then, we used the QWEN/QWEN2.5-32B-INSTRUCT-GPTQ-INT4 model to rate the relatedness of each query-text pair on a 5-point scale, producing token-level probabilities for scores "1" through "5".

**Training and Evaluation Details**

Table 2: Composition of the public portion of the E5 training dataset (Wang et al., 2023), reconstructed by Springer et al. (2024). In the task categories below, Natural Language Inference is denoted by NLI and Question-Answering as QA. Details on the instructions used for each dataset can be found at Section 6 of the Appendix.

| Dataset Name | Task Category | Meaning of anchor/associated text |
|---|---|---|
| AllNLI (Gao et al., 2021) | NLI | premise/hypothesis |
| DuReader (He et al., 2018) | Passage Retrieval | query/passage |
| ELI5 (Fan et al., 2019) | Popular Responses | forum question/user response |
| FEVER (Thorne et al., 2018) | Fact Checking | claim/document |
| HotpotQA (Yang et al., 2018) | Passage Retrieval for Multi-Hop QA | query/passage |
| Miracl (Zhang et al., 2023b) | Passage Retrieval | query/passage |
| MrTydi (Zhang et al., 2021) | Passage Retrieval | query/passage |
| MSMARCO (Bajaj et al., 2018) | Passage Retrieval | query/document or passage |
| Natural Questions (Kwiatkowski et al., 2019) | Passage Retrieval | query/Wikipedia article |
| Quora Duplicates (DataCanary et al., 2017) | Duplicates Classification | forum question/forum question |
| SQuAD (Rajpurkar et al., 2016) | Passage Retrieval | query/Wikipedia article |
| T2Ranking (Xie et al., 2023) | Passage Retrieval | query/web passage |
| TriviaQA (Joshi et al., 2017) | Passage Retrieval | query/Wikipedia article |

Table 3: Instructions used for datasets contained in the public portion of the E5 dataset and the Lightblue reranker distillation mixture (LB).

| Dataset | Instruction(s) |
|---|---|
| E5/NLI | Given a premise, retrieve a hypothesis that is entailed by the premise
Retrieve semantically similar text |
| E5/DuReader | Given a Chinese search query, retrieve web passages that answer the question |
| E5/ELI5 | Provided a user question, retrieve the highest voted answers on Reddit ELI5 forum |
| E5/FEVER | Given a claim, retrieve documents that support or refute the claim |
| E5/HotpotQA | Given a multi-hop question, retrieve documents that can help answer the question |
| E5/MIRACL | Given a question, retrieve Wikipedia passages that answer the question |
| E5/MrTyDi | Given a question, retrieve Wikipedia passages that answer the question |
| E5/MSMARCO Passage | Given a web search query, retrieve relevant passages that answer the query |
| E5/MSMARCO Document | Given a web search query, retrieve relevant documents that answer the query |
| E5/NQ | Given a question, retrieve Wikipedia passages that answer the question |
| E5/QuoraDuplicates | Given a question, retrieve questions that are semantically equivalent to the given question
Find questions that have the same meaning as the input question |
| E5/SQuAD | Retrieve Wikipedia passages that answer the question |
| E5/T2Ranking | Given a Chinese search query, retrieve web passages that answer the question |
| E5/TriviaQA | Retrieve Wikipedia passages that answer the question |
| LB/cpgqa | Given a question about clinical practice guidelines, retrieve relevant passages that answer the question |
| LB/logqa | Given a question about a system's log, retrieve the most relevant log entries |
| LB/lsat | Retrieve the passage which is most relevant to the given LSAT question |
| LB/narrativeqa | Retrieve the story that is most relevant to the given question |
| LB/pubmedqa | Given a biomedical research question, retrieve relevant passages that answer it |
| LB/qasports | Given a sports question, retrieve relevant passages that answer the question |

**Training Details.** We employ the Adam optimizer (Kingma & Ba, 2015) with $\beta_1 = 0.9$, $\beta_2 = 0.98$. We apply no weight decay. For the learning rate scheduler, we adopt a linear warmup over 5% of the total training steps, followed by a linear decay till end of training. We train all models for a constant of 4 epochs. The logit scale $\alpha$ is tuned and we find the value 20 to work well, which corresponds to a temperature of 0.05. In all experiments,

we make use of in-batch negatives, except for Margin MSE where we observed that the performance drops. We implement task-conditioned sampling for batching, in which all the samples in a batch belong to the same task. Doing so improves the quality of in-batch negatives as all the associated text belong to the same domain. The learning rate is initially set to a base batch size of 16 and then scaled according to the square root of the ratio between the total batch size across devices and the base batch size (Malladi et al., 2022), expressed as $\sqrt{\frac{total\_batch\_size}{16}}$. We apply gradient checkpointing. For our experiments, we use batch size 256 for all models, except some ablation where we use 128 for GPU economy. Context length is set to 8192 for all models, except 3B models where it is set at 4096 for memory economy. For similar reasons, we use LORA parameter efficient finetuning with 32 ranks for those models.

**Model Selection.** For model selection, we derive a validation score by using a subset of tasks from BEIR, which we call BEIR (short) throughout the paper. In particular, we take the DBPedia, HotpotQA, FiQA2018, FEVER, QuoraRetrieval and NQ tasks, and we subsample 256 queries and up to 131,072 passages from each of them. We do this because evaluating many retrieval tasks is time-consuming due to their large corpus size, often in the millions of documents. We periodically evaluate the performance of the sentence embedders that we train on the validation set, by computing the average NDCG@10 score over our validation tasks, and we select the best model that occurs during a training trial. Similarly, we use the same validation score to perform hyperparameter search and analysis.

**LB, E5, BEIR and MMTEB instructions.** As we have mentioned at Section 5, for some trials we use the public portion of the E5 dataset, as reproduced by Springer et al. (2024), for training our sentence embedders. In Table 2, we describe the composition of training sets and their corresponding tasks. In addition to comprising public permissible datasets, this training data mix has minimal overlap with MTEB tasks, enabling us to evaluate the robustness of our models on domains beyond the training set. Training encoders can leverage an extra conditioning signal to improve generalization across tasks (Su et al., 2023). This conditioning signals comes in the form of a task-specific instruction, which can be combined with the anchor sentence, and potentially with its associated sentences. In Section 6, we provide the list of instructions we used to augment the anchor texts, and also their associated text in the case of symmetric tasks. We use the same task instructions as previous works (Wang et al., 2023; Springer et al., 2024; BehnamGhader et al., 2024).

Similarly for the Lightblue (LB) distillation dataset (Lightblue, 2025), each row of text pairs originates from a multilingual QA dataset. The dataset and its composition is described in detail in Section 6. For some of the datasets contained, we define a task instruction that reflects the domain of the query contained in a row of the training dataset. If a specific task instruction has not been defined for a dataset, we fallback to a default one: "*Retrieve the most relevant passages to the given query*". In Section 6, we provide the list of instructions we used for some of the datasets in the Lightblue data mix.

During inference, we allow a user to provide an instruction of their liking. For benchmarks, like BEIR or MMTEB, we use the predefined set of task instruction provided by the software package for each of the tasks contained. Please refer to the MMTEB benchmark paper for more details (Enevoldsen et al., 2025). For TREC-DL evaluations, we use the instruction corresponding to MSMARCO: "Given a web search query, retrieve relevant passages that answer the query".

**System prompts for zero-shot QWEN2.5-32B-INSTRUCT rankers.** In Figure 2, we evaluate against zero-shot QWEN2.5-32B-INSTRUCT rankers in a 100kwqrel subset of TREC-DL 19-23 benchmark we have collected. For ranking with graded relevance (0-3), we apply the following system prompt: "*Your task is to judge how well the passage answers the query on a scale from 0 to 3. 0 - Irrelevant, 1 - Relevant topic, but does not contain the answer, 2 - Highly relevant, partial or unclear answer, and 3 - Perfectly relevant, exact answer. Answer with 0, 1, 2 or 3.*". For the binary relevance evaluation (0-1), we apply the following instruction: "*Your task is to judge how well the passage answers the query. 0 - Irrelevant, 1 - Perfectly relevant, exact answer. Answer with 0, or 1.*".

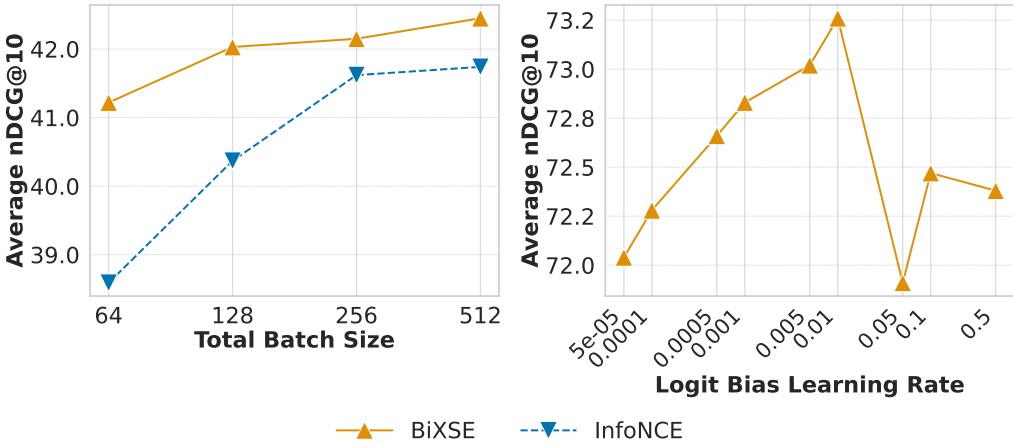

Figure 5: Batch size and logit bias lr

**Extended Study**

**BIXSE vs InfoNCE and weakly-supervised dataset noise.** We hypothesize that BCE achieves better robustness to weakly-supervised dataset noise, because the effect of noise is more diluted in the gradients of the BCE objective. Suppose the binary label case with hard negatives. At each training step the model encounters a batch of size $B$, the BCE-loss then is the average of losses from $2B^2$ binary classification predictions, while softmax-based loss is the average of losses from $B$ multi-class classification predictions. Assuming that the in-batch negatives are indeed true negatives: If one pair of texts in the batch is mislabeled, then for the softmax-loss this means that the $\frac{1}{B}$ predictions is optimized towards a false target, whereas for the BCE-loss $\frac{1}{B^2}$ predictions is optimized towards a false target.

**BIXSE vs InfoNCE and in-batch negatives.** We evaluate QWEN2.5-0.5B-INSTRUCT models by monitoring the average nDCG@10 on BEIR how performance changes as the total training batch size increases. For both InfoNCE and BIXSE, we use in-batch negatives, so larger batches provide more negative examples per query. As shown in Figure 5 (left), InfoNCE benefits noticeably from increasing batch size, consistent with its reliance on strong negative contrast. BIXSE, by contrast, achieves strong performance even at smaller batch sizes and shows only modest gains with larger ones. This suggests that BIXSE is less dependent on large quantities of in-batch negatives.

We can get intuition about the batch size resilience of BCE losses by analyzing loss gradients. In softmax-based contrastive learning, the denominator, in which the negative pairs appear, can be expressed as a log-sum-exp. This means that each of the $(x_k, y_n)$ pairs' gradient contribution is weighted by each pair's probability to have $y_n$ classified from $x_n$.

$$\frac{\partial}{\partial y_n} \log \sum_{m=1}^{M} \exp\left(s(x_k, y_m)\right) = \frac{\exp\left(s(x_k, y_n)\right)}{\sum_{m=1}^{M} \exp\left(s(x_k, y_m)\right)} \frac{\partial}{\partial y_n} s(x_k, y_n)$$

Combining this with large logit scales $\alpha \in (10, 100)$, that are used in order to achieve good downstream performance empirically, we can see that the negative pair with the largest logit $s(x_k, y_{m^*})$ contributes almost all of the gradient, while the rest of the negative pairs with smaller logits $s(x_k, y_m) \leq s(x_k, y_{m^*})$ have significantly smaller gradient contributions. In contrast, BCE loss weights all negative pairs independently from one another.

$$\frac{\partial}{\partial y_n} - \log \sigma\left(-s(x_k, y_n)\right) = \sigma\left(s(x_k, y_n)\right) \frac{\partial}{\partial y_n} s(x_k, y_k)$$

Combining this with large logit scales $\in (10, 100)$ once more, we can see that only the negative pairs that are perceived as positive by the model are going to be used and they will

| Benchmark | BIXSE (OURS) | PAIRWISE BCE | LAMBDA - NDCG V1 | LAMBDA - NDCG V2 |
|---|---|---|---|---|
| BEIR (short) | **73.91** | 72.17 | 72.43 | 73.78 |
| BEIR (full 15-task) | **45.05** | 43.30 | 41.40 | 44.68 |
| MTEB (Multilingual v1) | 55.46 | 53.50 | 53.46 | **55.62** |
| TREC (qrels 19–23) | **57.89** | 55.74 | 55.25 | 56.39 |
| TREC 2021 (top-100 docs) | **66.55** | 65.02 | 65.04 | 64.84 |
| TREC 2022 (top-100 docs) | 36.85 | 37.58 | **37.68** | 37.25 |
| TREC 2023 (top-100 docs) | 38.04 | 37.29 | **38.72** | 38.23 |

Table 9: NDCG@10 for LightBlue-trained models (in-batch only, batch size = 256) across standard retrieval benchmarks. Bold indicates the best score per benchmark.

have approximately equal weight ($\approx 1$) among themselves. We argue that this makes the BCE formulation use in-batch negatives more efficiently than softmax-based formulations, as it utilizes simultaneously all erroneous predictions about negative pairs instead of the most erroneous one at each training step.

**Effect of logit bias learning rate.** We further investigate the role of the logit bias $\beta$ by varying its learning rate while keeping the rest of the model fixed. As shown in Figure 5 (right) by the average nDCG@10 on BEIR (short), performance improves when the logit bias is trained with higher learning rates. For reference, we use a learning rate of $\approx 0.0001$ for QWEN2.5-0.5B-INSTRUCT base models at batch size 256. The results support our design choice: a fast-updating $\beta$ can quickly adapt to and absorb marginal label imbalance, particularly that induced by in-batch negatives, allowing the encoder to focus on modeling actual query-document relevance. Performance drops when $\beta$ is under-optimized, indicating that insufficient correction for label skew reduces the effectiveness of in-batch negatives.

**Analytical Report against InfoNCE**

**Comparative Evaluation Against Pairwise Ranking Objectives**

To assess BIXSE's competitiveness relative to traditional pairwise losses, we conducted extensive experiments comparing it to three strong baselines: Pairwise BCE (as used in RankNet (Burges et al., 2005) and PairDistill (Huang & Chen, 2024)), LambdaLoss (Wang et al., 2018) with two NDCG weighting variants, and MarginMSE (Hofstätter et al., 2021). These experiments were carried out on two diverse datasets, LightBlue (multilingual, in-batch only) and BGE-M3 (Chen et al., 2024) training datasets (English only subset, with hard negatives), using the QWEN2.5-0.5B-INSTRUCT model architecture. For fair comparison, we run hyperparameter search for all methods to tune for learning rates and logit scales, as well as hyperparameters specific to each training loss.

**LightBlue Results: Multilingual with LLM-graded relevance annotations** As shown in Table 1 and Table 9, BIXSE clearly outperforms InfoNCE and delivers stronger NDCG@10 scores than pairwise alternatives across all evaluation suites, including BEIR (short), BEIR (full), MTEB (Multilingual v1), and TREC-DL 2021–2023 (where we use the top-100 document lists provided by the official competitions). LightBlue contains only one graded document per query, so we make use of in-batch negatives for all training methods. We train for 1 epoch with batch size 256.

**BGE-M3 Results: English data with mined and scored hard-negatives** To investigate training efficiency under fixed compute and memory budgets, we evaluate model performance across configurations that varied the number of hard negatives and batch size. As we see in Tables 11 and 13, BIXSE benefits greatly from scaling batch size and leveraging in-batch negatives, while pairwise losses require multiple explicitly annotated negatives per query.

These results highlight the core advantage of BIXSE: performance improves with batch size without requiring additional annotated negatives. In contrast, pairwise methods depend more heavily on hard negatives and saturate early.

| MTEB Task | BIXSE (BCE) | PAIRWISE BCE | LAMBDA - NDCG V1 | LAMBDA - NDCG V2 |
|---|---|---|---|---|
| StackOverflowQA | 85.75 | 83.60 | 85.45 | 88.06 |
| TwitterHjerneRetrieval | 50.70 | 47.04 | 46.59 | 52.36 |
| AILAStatutes | 24.14 | 21.42 | 21.50 | 27.42 |
| ArguAna | 50.79 | 52.51 | 52.10 | 54.18 |
| HagridRetrieval | 98.90 | 98.75 | 98.50 | 98.75 |
| LegalBenchCorporateLobbying | 93.39 | 92.16 | 92.09 | 92.78 |
| LEMBPasskeyRetrieval | 84.75 | 79.25 | 84.00 | 84.50 |
| SCIDOCS | 16.12 | 17.71 | 16.61 | 17.42 |
| SpartQA | 6.05 | 8.15 | 3.92 | 11.77 |
| TempReasonL1 | 1.92 | 1.57 | 1.64 | 1.40 |
| TRECCOVID | 72.12 | 72.30 | 58.82 | 70.88 |
| WinoGrande | 42.77 | 43.02 | 45.39 | 40.51 |
| BelebeleRetrieval | 57.76 | 56.01 | 56.65 | 57.94 |
| MLQARetrieval | 70.68 | 66.84 | 68.35 | 69.64 |
| StatcanDialogueDatasetRetrieval | 23.73 | 21.29 | 23.38 | 25.29 |
| WikipediaRetrievalMultilingual | 86.34 | 79.68 | 85.42 | 84.48 |
| CovidRetrieval | 80.14 | 76.95 | 80.18 | 79.90 |
| MIRACLRetrievalHardNegatives | 52.30 | 44.91 | 41.80 | 47.01 |

Table 10: Analytic breakdown of NDCG@10 on multilingual MTEB tasks for LightBlue-trained models. Each column corresponds to a different training objective: BIXSE (BCE), PAIRWISE BCE, LAMBDALOSS (NDCG V1), and LAMBDALOSS (NDCG V2).

| # Hard Negatives / Batch Size | BIXSE (BCE) | PAIRWISE BCE | LAMBDALOSS (NDCG V2) |
|---|---|---|---|
| 15 / 16 | 43.63 | 41.68 | 49.84 |
| 7 / 32 | 44.95 | 44.04 | 50.55 |
| 3 / 64 | 46.29 | 45.82 | **51.82** |
| 1 / 128 | 45.92 | **51.06** | 50.84 |
| 0 / 256 | **48.88** | 45.66 | 45.49 |

Table 11: Summary of model performance (average NDCG@10) on the 15-task BEIR benchmark for models trained on the BGE dataset.

**Efficiency and Scalability Considerations**    BIXSE offers compelling annotation and training cost benefits:

- **Annotation Cost**: BIXSE achieves competitive performance using a single (graded) relevance label per query. Pairwise methods require at least four to eight supervised comparisons in order to perform optimally, increasing the cost when using expensive teacher models (e.g., LLMs or cross-encoders).

- **Training Cost**: BIXSE's pointwise BCE loss scales quadratically with batch size ($O(B^2)$), while pairwise losses scale cubically ($O(B^3)$) when using all document pairs. More importantly, the memory required to store pairwise score tensors grows rapidly and becomes impractical for large batches. For instance, storing score tensors for a batch of 16,384 using LambdaLoss exceeds 134 GB, while BIXSE only requires ≈134 MB.

We acknowledge the strength of LambdaLoss nDCG-v2 described by Wang et al. (2018). It performs close or better to BIXSE on several benchmarks, particularly when provided with ample labeled negatives. However, it lacks the same degree of annotation and memory efficiency. Importantly, BIXSE achieves similar or better downstream performance without compromising on scale.

| BEIR Task | BIXSE (BCE) | PAIRWISE BCE | LAMBDALOSS (NDCG V2) |
|---|---|---|---|
| TREC-COVID | 53.4 | 50.2 | 57.1 |
| BioASQ | 45.2 | 42.9 | 46.8 |
| NFCorpus | 38.1 | 36.5 | 40.2 |
| NQ | 49.8 | 48.7 | 52.6 |
| HotpotQA | 62.4 | 60.1 | 64.0 |
| FiQA-2018 | 34.9 | 35.5 | 38.4 |
| ArguAna | 30.6 | 28.9 | 34.1 |
| Touché-2020 | 28.3 | 26.8 | 31.7 |
| Quora | 89.3 | 91.2 | 90.8 |
| DBPedia | 42.6 | 41.3 | 45.1 |
| SCIDOCS | 18.4 | 17.7 | 20.5 |
| FEVER | 72.9 | 70.1 | 75.4 |
| Climate-FEVER | 24.7 | 23.9 | 27.3 |
| SciFact | 59.3 | 57.2 | 60.9 |
| CQADupStack | 32.1 | 31.6 | 34.0 |

Table 12: Analytic breakdown of model performance (NDCG@10) across individual tasks in BEIR. Results shown for the best-performing configuration per loss type among BGE-trained models.

| # Hard Negatives / Batch Size | BIXSE (BCE) | PAIRWISE BCE | LAMBDALOSS (NDCG V2) |
|---|---|---|---|
| 15 / 16 | 44.09 | 40.89 | 48.42 |
| 7 / 32 | 45.45 | 45.62 | 49.92 |
| 3 / 64 | 46.60 | 44.05 | **51.91** |
| 1 / 128 | 46.50 | **49.38** | 49.49 |
| 0 / 256 | **50.62** | 46.46 | 47.73 |

Table 13: Summary of model performance (average NDCG@10 score) across MTEB v2 English retrieval tasks) for BGE-trained models. Results are grouped by number of hard negatives and batch size.

| MTEB Task | BIXSE (BCE) | PAIRWISE BCE | LAMBDALOSS (NDCG V2) |
|---|---|---|---|
| ArguAna | 65.02 | 68.29 | 70.98 |
| CQADupstackGamingRetrieval | 54.32 | 54.47 | 54.09 |
| CQADupstackUnixRetrieval | 39.99 | 42.25 | 41.34 |
| ClimateFEVER | 26.88 | 23.86 | 27.06 |
| FEVER | 75.90 | 78.87 | 71.98 |
| FiQA2018 | 46.40 | 39.65 | 46.40 |
| HotpotQA | 64.42 | 61.92 | 64.42 |
| SCIDOCS | 22.24 | 19.86 | 22.24 |
| TRECCOVID | 73.29 | 74.73 | 76.04 |
| Touche2020Retrieval.v3 | 54.96 | 54.99 | 54.99 |

Table 14: Analytic breakdown of NDCG@10 for individual retrieval tasks in MTEB (English v2) for BGE-trained models. Each column reports the best-performing configuration for that loss.

