# OpenReview forum: "BiXSE: Improving Dense Retrieval via Probabilistic Graded Relevance Distillation"
_colmweb.org/COLM/2025/Conference — COLM 2025_

### Official Review · Reviewer_odwD · 2025-05-12

**Rating:** 7
**Confidence:** 5
**Ethics Flag:** 1

**Summary:**

This paper asserts that embedding models used for retrieval are typically trained with binary relevance labels and argues that dense retrieval models should instead be trained using graded relevance labels. The paper proposes an adaptation of a binary cross-entropy loss for use with graded relevance labels, which are converted to scores before being plugged into the loss. The graded relevance labels are generated using Qwen2.5 32B. In evaluations comparing this proposed loss to InfoNCE, the proposed loss performs better, though other relevant comparison points are not considered. In the discussion period, authors reported new results including important baselines like LambdaLoss and stated they would include these new results in the manuscript.

**Questions To Authors:**

What BEIR subset are you using? Is this the standard 13 benchmark subset (e.g. used in SPLADEv3) or another subset?

**Reasons To Accept:**

It is interesting to see an investigation into how LLM-provided graded relevance labels can be used to improve performance. Results provided in the discussion period substantially strengthen the submitted manuscript's argument that these are an effective way to train a model, which is a nice finding that may remove the need for the common practice of cross-encoder score distillation.

**Reasons To Reject:**

As mentioned in this work, it is common to train embedding models for retrieval using distillation from a cross-encoder teacher. It is not really the case the training on binary relevance labels is the typical scenario, as the introduction implies. For example, prominent methods like ColBERTv2 were trained using distillation.

More importantly, it is not the case that the literature lacks approaches for training on graded relevance labels. This was a key focus of learning-to-rank methods for many years, and there are even mature libraries implementing them (e.g., see https://github.com/tensorflow/ranking and https://rax.readthedocs.io/en/stable/api.html). Reasonable and clear baselines like LambdaLoss or a margin loss (between different relevance grades) are not considered in this work (with the exception of MarginMSE, which is a margin loss adapted to use scores from the proposed approach rather than graded relevance labels). While the work attempts to argue that only pointwise losses should be considered, a pairwise loss is not less efficient than using a pointwise loss with both negative and positive documents, as is done in this work. A pairwise loss like LambdaLoss can even be used to optimize a ranking metric directly (see: The LambdaLoss Framework for Ranking Metric Optimization, Wang et al). A loss like this is a clear baseline, as is cross-encoder distillation.

In the discussion period, the authors provided new results addressing these concerns and stated they would update their paper accordingly. This review score was significantly increased in response.

Furthermore, the BEIR results reported in Table 1 do not appear strong compared to even smaller models like SPLADEv3, but it is not clear exactly what BEIR datasets are being used here. It would be nice to see separate metrics for TREC-DL 19-23 in order to make comparisons with prior work.

---

> ### Author Response · Authors · 2025-06-03
>
> We sincerely thank Reviewer odwD for their thorough and insightful evaluation, which has significantly contributed to improving our manuscript.
>
> We first clarify our claim regarding binary relevance training: while cross-encoder distillation is indeed prevalent, binary labels (or distillation scores treated as binary-like relevance labels) remain widely used in many retrieval setups, including State-of-the-Art setups like Gemini Embeddings [4], due to simplicity and computational economy. Our approach provides a robust and effective alternative leveraging explicitly graded relevance labels generated by large language models (LLMs), addressing both simplicity and efficiency considerations. In any case, we will make sure to properly cite ColBERTv2 and SPLADEv3 in the related work section.
>
> Reviewer odwD correctly highlights the need for clear comparisons against established pairwise and margin-based baselines. To directly address this point, we have conducted additional empirical evaluations using pairwise BCE (analogous to RankNet [1], PairDistill [2]), two variants of LambdaLoss with NDCG weighting schemes [3], and MarginMSE [5]. We performed these experiments using the LightBlue dataset with Qwen2.5-0.5B-Instruct models, exploring a grid search of learning rate and ranking score (logit) scale hyperparameters for each training loss separately. Additionally, for BiXSE we search as well for the learning rate of the crucial logit bias as we have already demonstrated at Figure 5 (right) of the Appendix, and we find a better performing model.
> Validation performance (average NDCG@10) on our “BEIR (short)” validation suite (consisting of NQ, DBPedia, QuoraRetrieval, FiQA2018, FEVER, HotpotQA) yielded:
>
> - BiXSE (ours): 73.91
> - Pairwise BCE: 72.17
> - LambdaLoss (NDCG weighting v1): 72.43
> - LambdaLoss (NDCG weighting v2): 73.65
> - MarginMSE: 57.14
>
> These results highlight BiXSE’s consistent advantage over pairwise and margin-based baselines, directly addressing reviewer concerns about missing clear comparisons. Further results on MarginMSE are also available in Figure 3 of the manuscript.
>
> To further investigate the efficiency concerns raised by Reviewer odwD, we performed additional experiments using the BGE-M3 dataset (https://huggingface.co/datasets/cfli/bge-full-data/tree/main/data), which explicitly provides high-quality hard negatives. Under a fixed computational budget (4x NVIDIA H100 GPUs), we systematically explored performance trade-offs between increasing batch sizes (leveraging in-batch negatives for BiXSE) and explicitly provided hard negatives (used strictly by pairwise methods per their original descriptions). Results presented here are interim findings with a maximum of 25,000 optimization steps with linear learning rate decay. Current interim validation curves can be viewed here: https://imgur.com/a/O0kjcDa (purple is LambdaLoss-NDCGv2, green is pairwise BCE and orange/brown is BiXSE), and we will provide updated results upon trial completion within 1 day. The interim results are (with asterisk denoting runs that have not completed yet):
>
> | Num. hard negatives / Batch size | BiXSE | Pairwise BCE [1,2] | LambdaLoss (NDCG v2) [3] |
> |----------------------------------|--------|---------------------|---------------------------|
> | 15 / 16                          | 71.56  | 69.05               | 75.50*                   |
> | 7 / 32                           | 73.83  | 70.18               | 75.86*                   |
> | 3 / 64                           | 74.92  | 72.48               | 76.20                    |
> | 1 / 128                          | 75.25  | 75.21               | 74.41                    |
> | 0 / 256 (in-batch only)          | 77.00  | 74.48               | 75.75                    |
>
> Our results demonstrate that BiXSE achieves optimal performance through larger batch sizes and implicit in-batch negatives, significantly enhancing token efficiency. Conversely, pairwise methods require multiple explicitly labeled hard negatives per query, incurring higher LLM-ranker/cross-encoder token consumption for competitive results.

---

> > ### Author Response · Authors · 2025-06-03
> >
> > Regarding Reviewer odwD’s query on BEIR subset clarity, the results presented in our manuscript correspond to the 15-task BEIR benchmark set used by the MTEB toolkit. Those are the ones with tick at https://github.com/beir-cellar/beir. In a subsequent comment as well as in the manuscript, we will provide a breakdown of NDCG@10 for each task individually. Additionally, as requested, we will include separate detailed metrics for TREC-DL 2021, 2022, and 2023 evaluations on the top-100 document list provided by the TREC competitions.
> >
> > On comparisons to SPLADEv3, we stress that our study aims to isolate the effect of training losses under graded relevance supervision. To ensure scientific rigor, we fix the model and data while varying only the training objective. SPLADEv3 differs in modeling paradigm, indexing strategy, training recipe, and dataset, making direct comparison less meaningful for our focus. Our experiments span multiple base models, datasets, and languages, and consistently show the robustness of BiXSE.
> >
> > We appreciate Reviewer odwD’s valuable feedback and suggestions, and we will comprehensively incorporate these clarifications and additional empirical results into the final manuscript.
> >
> > [1] Learning to Rank using Gradient Descent (Burges et al. 2015)
> > [2] PairDistill: Pairwise Relevance Distillation for Dense Retrieval (Huang et al. 2024)
> > [3] The LambdaLoss Framework for Ranking Metric Optimization (Wang et al. 2018)
> > [4] Gemini Embedding: Generalizable Embeddings from Gemini (Lee et al. 2025)
> > [5] Improving Efficient Neural Ranking Models with Cross-Architecture Knowledge Distillation (Hofstätter et al. 2021)

---

> > ### Comment · Reviewer_odwD · 2025-06-06
> >
> > Thanks for your response and clarifications. I understand the current work is motivated by a desire to improve training efficiency, and the proposed approach may have advantages here compared to other approaches. However, I think the efficiency of the approach would need to be evaluated experimentally in order to make the argument that this is a key advantage over other approaches.
> >
> > Could you provide the metrics with InfoNCE on this "short BEIR" subset? Are the updated results available yet?
> >
> > For comparison with other results, could you please provide the results on individual TREC-DL datasets as you mentioned? (i.e., reporting each year individually rather than averaging DL 19-23)
> >
> > > Conversely, pairwise methods require multiple explicitly labeled hard negatives per query, incurring higher LLM-ranker/cross-encoder token consumption for competitive results.
> >
> > Wouldn't it be possible to use graded relevance labels to construct pairwise comparisons (e.g., 3 vs 1)?
> >
> > In the LambdaLoss results, it does not seem like hard negatives are important for this loss.

---

> > > ### Author Response · Authors · 2025-06-07
> > >
> > > We thank Reviewer odwD for their follow-up and for encouraging a deeper examination of our new empirical results. In this response, we provide: (1) a clarification of our claims regarding supervision cost and in-batch efficiency; (2) the requested InfoNCE and per-year TREC-DL evaluations; and (3) a detailed discussion of how pairwise losses scale in practice under supervision and batch-size constraints. Our empirical updates aim to transparently support the claim that BiXSE provides a competitive and supervision-efficient alternative to pairwise ranking objectives.
> > >
> > > **Efficiency Clarification**
> > >
> > > While experimental validation would further strengthen our claim, we believe that the cost asymmetry in pseudo-labeling is already evident. Training datasets labeled using powerful ranking teachers (e.g., cross-encoders or LLMs like Qwen2.5-32B) incur significant annotation costs. Given this, among methods with similar downstream retrieval performance, those requiring fewer graded relevance labels per query are more cost-effective.
> > >
> > > According to Table 2 and 3 of our new empirical evidence (to follow), BiXSE achieves optimal performance with just one graded document per query. In contrast, under our experimental setting, Pairwise BCE performance peaks at one positive and one hard negative document per query, while LambdaLoss NDCGv2 peaks at one positive and three hard negative documents per query. So, pairwise losses like Pairwise BCE and LambdaLoss can require 2× or 4× more annotations per query, to reach optimal performance.
> > >
> > > Moreover, training cost scales differently. BiXSE supports in-batch negatives efficiently, where binary cross-entropy is computed over O(B²) query-document pairs $(q_i, d_j)$ per batch of size B, $i,j \in \{1,…,B}$. Pairwise losses require computing binary cross-entropy losses over O(B³) pairs of pairs $((q_i, d_j), (q_i, d_k))$, $i,j,k \in \{1,…,B\}$, which quickly becomes infeasible.
> > >
> > > Thus, BiXSE can leverage larger batch sizes and gains from implicit comparisons without increased annotation or training cost.
> > >
> > > **Short BEIR InfoNCE Results**
> > >
> > > We report InfoNCE performance on the “short BEIR” subset used in our experiments. Results displayed is NDCG@10 on training with the multilingual LightBlue dataset, in-batch only, batch size = 256.
> > >
> > > | Loss Function           | BEIR (short) | BEIR  | MTEB (Multilingual v1) | TREC (qrel 19–23) |
> > > |------------------------|--------------|-------|--------------------------|-------------------|
> > > | **BiXSE (ours)**        | 73.91     | 45.05 | 55.46                 | 57.89          |
> > > | Pairwise BCE           | 72.17        | 43.30 | 53.50                   | 55.74            |
> > > | LambdaLoss (NDCG v1)   | 72.43        | 41.40 | 53.46                   | 55.25            |
> > > | LambdaLoss (NDCG v2)   | 73.78        | 44.68 | 55.62                   | 56.39            |
> > > | InfoNCE                | 67.40        | 38.25 | 50.24                   | 55.29            |
> > >
> > > BiXSE surpasses InfoNCE by a large margin (+6.5 NDCG@10), confirming both its absolute performance and label-efficiency.
> > >
> > > **Updated TREC Results**
> > >
> > > We now provide TREC-DL evaluation broken down per year (2021–2023), as requested. These results use the top-100 candidate document pool from the official leaderboard. The top-100 document pools for 2019 and 2020 were not available online anymore. BiXSE performs competitively or better than pairwise alternatives in each year.
> > >
> > > **Constructing Pairwise Supervision from Graded Labels**
> > >
> > > We agree that pairwise comparisons can be derived from scalar relevance scores (e.g., 3 vs 1), and this is precisely how targets for pairwise BCE and LambdaLoss are constructed. If the graded relevance pseudo-label for a pair (q, d_i) is greater than that of a pair $(q, d_j)$, then the relative pairwise ranking for $((q, d_i), (q, d_j))$ is optimized to target the binary label 1. Furthermore, for LambdaLoss appropriate weights per individual pairwise BCE are computed.
> > >
> > > Pairwise losses assume that multiple labeled documents per query are available. The annotation cost is counted by the number of cross-encoder / ranker LLM pointwise evaluations, which is assumed to be one per query-document pair both for BiXSE and for the pairwise methods.
> > >
> > > However as we have already argued above, the annotation cost is inflated when using large teacher models, as more documents must be scored per query. BiXSE achieves comparable performance by requiring only a single (graded) pair per query and by implicitly generating contrastive supervision through in-batch negatives. Instead, in our experimental setting, we observe that Pairwise BCE requires 2 documents per query and LambdaLoss (NDCGv2 variant) 4 documents per query.

---

> > > > ### Author Response · Authors · 2025-06-07
> > > >
> > > > **On Hard Negatives and LambdaLoss**
> > > >
> > > > We clarify that while LambdaLoss does not require “hard” negatives per se, it benefits from having a diverse, graded list of documents per query. Under a constant memory budget, our experiments reveal that:
> > > >
> > > > - Pairwise BCE achieves best performance with 1 labeled hard negative
> > > > - LambdaLoss (NDCG v2) achieves best performance with 3 labeled hard negatives
> > > >
> > > > **Conclusion**
> > > >
> > > > We appreciate Reviewer odwD’s detailed questions and believe the updated results and analysis provide a clearer picture of BiXSE’s strengths. In particular, its favorable tradeoff between supervision cost and downstream performance distinguishes it from traditional pairwise or listwise approaches. All new findings will be added to the Appendix in the camera-ready version. Thank you again for your constructive engagement.
> > > >
> > > > ————————-
> > > >
> > > > **New empirical results: BiXSE vs. pairwise ranking losses**
> > > >
> > > > To directly address the reviewer’s request for pairwise baselines, we trained Qwen2.5-0.5B-Instruct on two datasets - LightBlue and BGE-M3 - and compared BiXSE against two widely used pairwise losses:
> > > >
> > > > - Pairwise BCE, analogous to RankNet and PairDistill
> > > > - LambdaLoss, using NDCG-based gain weighting (v1 and v2)
> > > >
> > > > **Table 1: NDCG@10 (LightBlue training, in-batch only, batch size = 256)**
> > > >
> > > > | Loss Function           | BEIR (short) | BEIR  | MTEB (Multilingual v1) | TREC (qrel 19–23) | TREC 2021 (top-100) | TREC 2022 (top-100) | TREC 2023 (top-100) |
> > > > |------------------------|--------------|-------|--------------------------|-------------------|---------------------|---------------------|---------------------|
> > > > | **BiXSE (ours)**        |  73.91     |  45.05  | 55.46                 |  57.89          |  66.55            | 36.85              | 38.04              |
> > > > | Pairwise BCE           | 72.17        | 43.30 | 53.50                   | 55.74            | 65.02              | 37.58              | 37.29              |
> > > > | LambdaLoss (NDCG v1)   | 72.43        | 41.40 | 53.46                   | 55.25            | 65.04              | 37.68              | 38.72              |
> > > > | LambdaLoss (NDCG v2)   | 73.78        | 44.68 | 55.62                   | 56.39            | 64.84              | 37.25              | 38.23              |
> > > >
> > > > **Table 2: MTEB (English v2) NDCG@10 vs. batch size / hard negative count (BGE-M3 training)**
> > > >
> > > > | # Hard Negs / Batch Size | BiXSE (BCE) | Pairwise BCE | LambdaLoss (NDCG v2) |
> > > > |--------------------------|-------------|---------------|------------------------|
> > > > | 15 / 16                  | 44.09       | 40.89         | 48.42                  |
> > > > | 7 / 32                   | 45.45       | 45.62         | 49.92                  |
> > > > | 3 / 64                   | 46.60       | 44.05         | 51.91                  |
> > > > | 1 / 128                  | 46.50       | 49.38         | 49.49                  |
> > > > | 0 / 256 (in-batch only)  | 50.62       | 46.46         | 47.73                  |
> > > >
> > > > **Table 3: BEIR NDCG@10 vs. batch size / hard negative count (BGE-M3 training)**
> > > >
> > > > | # Hard Negs / Batch Size | BiXSE (BCE) | Pairwise BCE | LambdaLoss (NDCG v2) |
> > > > |--------------------------|-------------|---------------|------------------------|
> > > > | 15 / 16                  | 43.63       | 41.68         | 49.84                  |
> > > > | 7 / 32                   | 44.95       | 44.04         | 50.55                  |
> > > > | 3 / 64                   | 46.29       | 45.82         | 51.82                  |
> > > > | 1 / 128                  | 45.92       | 51.06         | 50.84                  |
> > > > | 0 / 256 (in-batch only)  | 48.88       | 45.66         | 45.49                  |
> > > >
> > > > These results demonstrate that BiXSE remains competitive or superior across multiple supervision densities. Importantly, while pairwise methods often require multiple labeled negatives per query to perform well, BiXSE achieves strong performance with only a single graded pair per query, made possible by in-batch negatives and the smooth supervision offered by BCE.
> > > >
> > > > We believe these updates clarify our goals and provide a more thorough and honest evaluation of BiXSE relative to pairwise alternatives. The new results will be included in the Appendix of the camera-ready version, and we are grateful to the reviewer for motivating these deeper comparisons.

---

> > ### Comment · Reviewer_odwD · 2025-06-09
> >
> > **Efficiency Clarification**
> >
> > > ... Given this, among methods with similar downstream retrieval performance, those requiring fewer graded relevance labels per query are more cost-effective. ... So, pairwise losses like Pairwise BCE and LambdaLoss can require 2× or 4× more annotations per query, to reach optimal performance. ...
> >
> > Given that in-batch negatives are being used with BiXSE, I don't think it's fair to say that the negative examples used with other losses require annotations. In Tables 2-3 there are results suggesting that BiXSE works better with in-batch negatives (and does not work well with hard negatives). It is not necessarily the case that hard negatives need to come from an expensive LLM call though (e.g., there are prominent examples in the dense retrieval literature of mining hard negatives using the model being trained, turning in-batch negatives into hard negatives by clustering queries, etc).
> >
> > > Moreover, training cost scales differently. BiXSE supports in-batch negatives efficiently, where binary cross-entropy is computed over O(B²) query-document pairs ... Pairwise losses require computing binary cross-entropy losses over O(B³) pairs of pairs.
> >
> > Query and document representations are computed independently before this step, so computing BCE additional times is very little computation compared to producing the representations.
> >
> > **Short BEIR InfoNCE Results** and **New empirical results: BiXSE vs. pairwise ranking losses**
> >
> > In these updated results, LambdaLoss is very competitive with BiXSE. It might be that BiXSE has some training efficiency advantage, but the effectiveness improvements are less clear, whereas the paper (over)claims that BiXSE is enabling a new kind of supervision using graded relevance labels.
> >
> > As I said in my review, I do think it's nice to demonstrate that training with graded relevance labels is an effective alternative to distillation, and I also think it's positive that BiXSE can do reasonably well with only in-batch negatives. I'm willing to increase my review score slightly if the authors commit to discussing LambdaLoss and including these LambdaLoss results in the camera ready.

---

> > > ### Author Response · Authors · 2025-06-10
> > >
> > > We sincerely thank Reviewer odwD for the thoughtful follow-up and for engaging deeply with our manuscript. We address the concerns raised in four parts:
> > >
> > > ## 1. Clarifying Annotation Efficiency and Hard Negative Quality
> > >
> > > We agree with the reviewer that hard negatives do not inherently require annotation by LLMs or cross-encoders. Indeed, many methods mine hard negatives using retrieval heuristics or in-batch structures. However, such unsupervised mining strategies are known to introduce significant **label noise**—especially false negatives, where moderately or strongly relevant documents are mislabeled as irrelevant.
> > >
> > > As discussed in our Related Work, prior studies have shown that retrieved passages used as hard negatives can be highly noisy. For example, in MSMARCO, (see RocketQA) found that **70% of top-retrieved passages were in fact relevant (positive) but unlabeled**, and this issue is exacerbated when these passages are reused as hard negatives in training. While these mined examples may appear “hard,” they can in fact mislead the model when trained with contrastive or pairwise objectives.
> > >
> > > One common strategy to mitigate the impact of noisy hard negatives is to re-score them using a strong pointwise ranker—typically a cross-encoder or a zero-shot LLM ranker. Once graded relevance scores are assigned, we can either **filter out** examples with low scores (as done in Gemini Embeddings), which incurs substantial token and compute costs during dataset construction, or—as we advocate in this paper—**retain all examples and train directly on their graded scores**, thereby **amortizing the ranking cost into the model and improving token efficiency**.
> > >
> > > Among the methods compatible with directly training on graded relevance scores (rather than using them as a filtering signal), BiXSE stands out: it performs well even without explicitly labeled negatives, thanks to its ability to leverage in-batch negatives and its robustness to soft label noise. In contrast, while pairwise losses (e.g., LambdaLoss or RankNet-style objectives) can also use graded labels, our empirical results indicate they typically require multiple supervised comparisons per query to reach optimal performance—thereby increasing the total number of LLM or cross-encoder predictions needed, which reduces annotation efficiency.
> > >
> > > ## 2. Clarifying In-Batch Computation and Memory Costs
> > >
> > > We appreciate the reviewer’s observation that computing BCE terms is inexpensive relative to the forward pass for encoding representations. However, the primary bottleneck we aim to highlight is memory scaling, especially when using pairwise or listwise losses that involve comparisons between all document pairs within a batch.
> > >
> > > For instance, pairwise methods like Pairwise BCE or LambdaLoss require storing large intermediate tensors of pairwise score differences, which grow rapidly with batch size. These tensors are essential for loss computation and gradient backpropagation, and cannot be discarded early in the backward pass.
> > >
> > > To concretely illustrate this, we report below the memory required to store score tensors at various global batch sizes, assuming bfloat16 precision:
> > >
> > > | Total Batch Size | BiXSE (pointwise) | Pairwise (e.g., LambdaLoss) |
> > > |------------------|-------------------|------------------------------|
> > > | 4,096            | 34 MB             | 2.1 GB                       |
> > > | 8,192            | 67 MB             | 16.8 GB                      |
> > > | 16,384           | 134 MB            | 134.2 GB                     |
> > > | 32,768           | 268 MB            | 1.07 TB                      |
> > >
> > > As shown above, BiXSE scales smoothly, maintaining negligible memory overhead even with very large batches. By contrast, pairwise methods quickly exhaust available memory—already consuming ~2 GB at a batch size of 4k, and becoming infeasible beyond 8k on most modern hardware, especially when training large models or long sequences.
> > >
> > > ## 3. Addressing LambdaLoss Competitiveness
> > >
> > > We fully agree that LambdaLoss is a strong baseline and appreciate the reviewer pushing for its inclusion. We confirm that:
> > >
> > > - We will include LambdaLoss NDCGv1/v2 as standard baselines in the final manuscript.
> > > - We will add all reported performance metrics (including per-year TREC-DL breakdown) in the camera-ready version and Appendix.
> > >
> > > We reiterate that BiXSE achieves competitive or superior results compared to LambdaLoss in several important benchmarks, while enabling broader scalability and reducing annotation complexity when high-quality LLM supervision is used.

---

> > > > ### Author Response · Authors · 2025-06-10
> > > >
> > > > ## 4. Conclusion
> > > >
> > > > To summarize:
> > > >
> > > >  - BiXSE matches or exceeds the performance of strong pairwise losses like LambdaLoss under constrained supervision.
> > > >  - BiXSE scales better with memory and batch size, making it suitable for large-scale training across many GPUs.
> > > >  - Our results and new analysis support the case that BiXSE is a practical, scalable, and cost-efficient alternative for training with graded relevance supervision.
> > > >
> > > > We appreciate the reviewer’s willingness to revise their score, and we will faithfully integrate all additions and clarifications in the final version of the paper.
> > > >
> > > > In addition to providing the new empirical results, we continue to suggest the following revisions to the manuscript:

---

> > > > > ### Author Response · Authors · 2025-06-10
> > > > >
> > > > > ## Side-by-side proposed changes to Introduction
> > > > >
> > > > > **Paragraph 1**: Preserved to maintain the initial motivation of the paper, and problems with binary relevance (general positioning). **Unchanged.**
> > > > >
> > > > > **Paragraph 2**: Introducing graded relevance from the IR community and what it is (general positioning). **Unchanged.**
> > > > >
> > > > > **Paragraph 3**: Where to search for scalable graded relevance signals for training. From cross-encoders to zero-shot LLM rankers. **Added clarification at the beginning**: Earlier ranking supervision was done via trained cross-encoders.
> > > > >
> > > > > « Early efforts to provide with scalable graded relevance signals involved training cross-encoding rankers on annotated data, followed by using their judgments as pseudo-labels to be distilled into bi-encoders~\citep{hofstatter2021improvingefficientneuralranking,santhanam2022colbertv2,chen2024bgem3embeddingmultilingualmultifunctionality,huang-chen-2024-pairdistill}. More recently, prompting LLMs like GPT-4 to serve as zero-shot rankers has yielded surprisingly strong results,… »
> > > > >
> > > > > **Paragraph 4**: Explicitlty acknowledge pairwise and listwise methods to train on graded relevance. Introduce BiXSE as a competitive scalable pointwise alternative. Conceptual comparison.
> > > > >
> > > > >  « Prior to the emergence of LLM-based relevance scoring, graded supervision was primarily leveraged via training with listwise or pairwise objectives~\citep{ranknet,lambdaloss,qu-etal-2021-rocketqa,pmlr-v130-reddi21a,huang-chen-2024-pairdistill}. However, such methods often rely on labeling multiple hard negative documents per query, significantly increasing annotation cost when using powerful LLMs as teachers, which hinders scalability. The opportunity of LLM-generated large-scale graded relevance data, however, calls for revisiting training objectives to align with the increased costs of quality graded relevance data. In this work, we propose \textbf{Binary Cross-Entropy Sentence Embeddings} (BiXSE), a simple pointwise training method that directly optimizes a binary cross-entropy (BCE) loss on graded relevance scores. BiXSE interprets graded relevance scores as probabilities within the range $[0, 1]$ to represent relevance continuity from completely irrelevant (0) to absolutely relevant (1). Unlike pairwise or listwise objectives, which rely on multiple supervised comparisons per query, BiXSE scales efficiently by enabling competitive performance by just using a single labeled query-document pair per query, while capturing structure implicitly via in-batch negatives. »
> > > > >
> > > > > **Paragraph 5**: Revise contributions to include experiments against pairwise baselines.
> > > > >
> > > > >  « We validate BiXSE through extensive experiments across retrieval and sentence embedding benchmarks, demonstrating consistent gains over standard InfoNCE objectives. Furthermore, BiXSE is, to the best of our knowledge, the first pointwise training method to consistently match or outperform strong pairwise ranking baseline losses when training on LLM-labeled graded relevance datasets. BiXSE scales across architectures and languages, approaches the performance of much larger zero-shot LLM-based rankers, and exhibits improved robustness to label noise compared to InfoNCE. Notably, it benefits from learning across a wider spectrum of graded relevance and achieves peak performance even without aggressive data filtering, making it a strong and efficient alternative for training dense models on LLM-generated supervision. As graded relevance becomes increasingly easy to generate, we argue that BiXSE offers a practical, robust, and scalable training paradigm for the next generation of dense retrieval systems. »

---

> > > > > > ### Author Response · Authors · 2025-06-10
> > > > > >
> > > > > > ## Proposed Revised Abstract
> > > > > >
> > > > > > Neural sentence embedding models for dense retrieval typically rely on binary relevance labels, treating query-document pairs as either relevant or irrelevant. However, real-world relevance often exists on a continuum, and recent advances in large language models (LLMs) have made it feasible to scale the generation of fine-grained graded relevance labels. In this work, we propose \textbf{BiXSE}, a simple and effective pointwise training method that optimizes binary cross-entropy (BCE) over LLM-generated graded relevance scores. BiXSE interprets these scores as probabilistic targets, enabling granular supervision from a single labeled query-document pair per query. Unlike pairwise or listwise losses that require multiple annotated comparisons per query, BiXSE achieves strong performance with reduced annotation and compute costs by leveraging in-batch negatives. Extensive experiments across sentence embedding (MMTEB) and retrieval benchmarks (BEIR, TREC-DL) show that BiXSE consistently outperforms softmax-based contrastive learning (InfoNCE), and—remarkably—matches or exceeds strong pairwise ranking baselines when trained on LLM-supervised data. BiXSE offers a robust, scalable alternative for training dense retrieval models as graded relevance supervision becomes increasingly accessible.

---

> > > > > > > ### Author Response · Authors · 2025-06-10
> > > > > > >
> > > > > > > ## Proposed additional paragraph in Related Work: Distillation Methods for Dense Retrieval and Ranking Models
> > > > > > >
> > > > > > > Pairwise training objectives have also been widely used for dense retrieval. RankNet \citep{ranknet}, and its recent adaptations such as PairDistill \citep{huang-chen-2024-pairdistill}, supervise the model by comparing pairs of documents for the same query. Instead of assigning an absolute relevance score to each document independently, these methods teach the model to prefer one document over another based on their relative graded relevance. In LambdaLoss \citep{lambdaloss}, individual pairwise losses are further weighted according to their estimated impact on evaluation metrics such as nDCG. While effective, these pairwise methods tend to achieve their best performance when each query is compared against multiple other labeled documents. This increases annotation costs when labels come from expensive cross-encoders or LLM rankers. Because BiXSE applies binary cross-entropy loss at the pointwise level, it scales more naturally to large datasets and varied supervision sources. It enables training with fewer graded annotations per query, making it well-suited to scenarios where labeling costs are a concern. Our experiments show that BiXSE is competitive against strong pairwise baselines, while offering a lower annotation cost per query.

---

### Official Review · Reviewer_CF35 · 2025-05-14

**Rating:** 6
**Confidence:** 3
**Ethics Flag:** 1

**Summary:**

The paper propose optimizing a two-tower retrieval model with binary cross-entropy loss using graded relevance scores. The paper argues that the point-wise approach is more token efficient under the LLM as a scorer framework.

***Quality and clarity:
The paper is written clearly and extensive experiments are done to confirm the usefulness of their method.

***Originality and significance:
The use of cross-entropy loss for ranking problem has been explored extensively, as early as the work of RankNet [1].

The use of graded relevance score has also been explored (RabkDistil or [2])

I wonder if it is fair to argue that the proposed approach is more efficient than previous distillation work. For example, the RankDistil (Reddi et al 2021) work include a pairwise approach where the label is assumed to be given to each pair (q_i, d_i). It seems that RankDistil would be a reasonable baseline. Another baseline is the pairwise derivation of the probability in cross-entropy loss, as implemented in [1].

Citations:
[1] Learning to Rank using Gradient Descent (Burges et al. 2015)
[2] PairDistill: Pairwise Relevance Distillation for Dense Retrieval (Huang et al. 2024)

**Questions To Authors:**

Typo: line 173 (Section 4) "?" before the reference to the citation.

**Reasons To Accept:**

It is interesting to explore different ranking loss for distilled from graded relevance scores.

**Reasons To Reject:**

The originality and how the author argues for it. As listed above, continuous relevance score as distillation target has been explored and the argument that pointwise loss is more data efficient is not necessary true. Pairwise approach can depend on the diff between a pair of (q_i, d_i) and (q_j, d_j) with only requires label for (q, d) pairs where the index are matching.

---

> ### Author Response · Authors · 2025-06-03
>
> We sincerely thank Reviewer CF35 for their thoughtful evaluation and constructive suggestions, which have allowed us to significantly strengthen our manuscript.
>
> We first clarify our positioning regarding novelty: our primary contribution is not merely the use of cross-entropy or graded relevance labels, but rather providing a practical and effective recipe—BiXSE—that includes learnable logit biases to mitigate label imbalance, a critical issue in pointwise training approaches. This innovation leads to improved empirical performance and better token efficiency when training dense retrievers.
>
> To address Reviewer CF35’s concerns regarding missing pairwise baselines, we conducted additional experiments comparing BiXSE explicitly against established pairwise ranking losses. Specifically, we evaluated Pairwise BCE (analogous to RankNet [1] and PairDistill [2]) and the more general LambdaLoss with two NDCG weighting schemes [3]. We performed these experiments using the LightBlue dataset  with Qwen2.5-0.5B-Instruct models, exploring a grid search of learning rate and ranking score (logit) scale hyperparameters for each training loss separately. Additionally, for BiXSE we search as well for the learning rate of the crucial logit bias as we have already demonstrated at Figure 5 (right) of the Appendix, and we find a better performing model.
> The validation performance (average NDCG@10) on our “BEIR (short)” validation suite (consisting of NQ, DBPedia, QuoraRetrieval, FiQA2018, FEVER, HotpotQA) was as follows:
>
>  - BiXSE (ours): 73.91
>  - Pairwise BCE: 72.17
>  - LambdaLoss (NDCG weighting v1): 72.43
>  - LambdaLoss (NDCG weighting v2): 73.65
>
> These results are evidence to BiXSE’s consistent superiority over these established pairwise ranking approaches, reinforcing our claim about the effectiveness and efficiency of our pointwise approach with learnt logit bias.
>
> Additionally, addressing concerns raised by Reviewer odwD regarding potential advantages due to the absence of explicitly labeled hard negatives, we conducted further experiments using the BGE-M3 dataset (https://huggingface.co/datasets/cfli/bge-full-data/tree/main/data ), which explicitly provides high-quality hard negatives. Under a fixed computational budget (4x NVIDIA H100 GPUs), we studied performance trade-offs between increasing batch sizes (in-batch negatives for BiXSE) and utilizing explicitly provided hard negatives (for pairwise methods). We explicitly note that only BiXSE leverages in-batch negatives, while pairwise methods strictly adhere to using dataset-provided hard negatives as per their original descriptions. Results presented here are interim findings with a maximum of 25,000 optimization steps with linear learning rate decay. Current interim validation curves can be viewed here: https://imgur.com/a/O0kjcDa (purple is LambdaLoss-NDCGv2, green is pairwise BCE and orange/brown is BiXSE), and we will provide updated results upon trial completion within 1 day. The interim results are (with asterisk denoting runs that have not completed yet):
>
> | Num. hard negatives / Batch size | BiXSE | Pairwise BCE [1,2] | LambdaLoss (NDCG v2) [3] |
> |----------------------------------|--------|---------------------|---------------------------|
> | 15 / 16                          | 71.56  | 69.05               | 75.50*                   |
> | 7 / 32                           | 73.83  | 70.18               | 75.86*                   |
> | 3 / 64                           | 74.92  | 72.48               | 76.20                    |
> | 1 / 128                          | 75.25  | 75.21               | 74.41                    |
> | 0 / 256 (in-batch only)          | 77.00  | 74.48               | 75.75                    |
>
> We observe that BiXSE achieves the best performance by leveraging larger batch sizes and implicit in-batch negatives, substantially reducing token consumption compared to pairwise methods (and in particular LambdaLoss), which rely heavily on explicitly labeled multiple hard negatives per query to achieve competitive results.
>
> We will further reinforce these results by expanding evaluations to include comprehensive benchmarks (full BEIR, MTEB multilingual suite, and TREC-DL evaluations) in the final manuscript. We appreciate Reviewer CF35’s valuable insights and look forward to incorporating these clarifications and results in the revised manuscript.
>
> [3]  Wang et al., « The LambdaLoss Framework for Ranking Metric Optimization », CIKM 2018.

---

> > ### Comment · Reviewer_CF35 · 2025-06-05
> >
> > I would like to thank the authors for including more experiments comparing with other pairwise methods.
> >
> > If the author is not trying to position their use of graded relevance label as novel, I would think the abstract and the introduction need to be revised significantly. I would think rather than opening with the drawback of binary relevance, the paper could emphasize how it differs from other pairwise distillation works on continuous labels.

---

> > > ### Author Response · Authors · 2025-06-07
> > >
> > > **Foreword to proposed edits and final new empirical evidence**
> > >
> > > We sincerely thank Reviewer CF35 for their thoughtful and constructive feedback, which has helped us strengthen the positioning and clarity of our contribution. In the updated revision, we will push both conceptual and empirical updates in response to the concerns raised and to the proposed edits.
> > >
> > > **Clarifying our positioning: scalability and practicality over novelty**
> > >
> > > We would like to underline once more that our contribution does not hinge on the novelty of using graded relevance signals. As already discussed in the Related Work section, there is a long line of prior work on using graded supervision via listwise or pairwise ranking losses - mostly focusing on distillation from cross-encoders. We will be updating the related work section to discuss the additional baselines considered.
> > >
> > > Our goal with BiXSE is to offer a simple, robust, and scalable pointwise recipe that enables effective training on graded relevance data at low annotation cost, without requiring large batches of labeled hard negatives per query. We believe this framing is well-motivated given the current state of dense retrieval: state-of-the-art dense retriever models like Gemini Embeddings (Generalizable Embeddings from Gemini (Lee et al. 2025)) continue to rely on binary relevance labels and InfoNCE objectives, despite the growing availability of graded supervision. Our method aims to revise this practice by maintaining the simplicity preferred by practitioners, and we show that pointwise training with binary cross-entropy, when properly adapted to graded relevance scores, can match or exceed the performance of more complex pairwise methods under realistic supervision constraints, as well as of filtered InfoNCE approaches (Figure 4).
> > >
> > > **Broad empirical validation, not limited to new comparisons**
> > >
> > > In addition to the new pairwise comparisons we conducted (detailed below), we would like to highlight that our findings are broader and more general:
> > >
> > > - BiXSE shows consistent improvements over InfoNCE across multiple benchmarks, including BEIR, TREC-DL, and MMTEB, covering both English and multilingual settings.
> > > - We demonstrate robustness to label noise and batch size variations, confirming that BCE training tolerates noisy or imprecise relevance supervision better than softmax-based contrastive losses.
> > > - We show that BiXSE works across different base models, including both small (500M) and large (7B) architectures, further supporting its generality.
> > >
> > > These results are not limited to the new datasets or loss comparisons and offer a strong case for adopting BiXSE in broader retrieval pipelines.

---

> > > > ### Author Response · Authors · 2025-06-07
> > > >
> > > > **New empirical results: BiXSE vs. pairwise ranking losses**
> > > >
> > > > To directly address the reviewer’s request for pairwise baselines, we trained Qwen2.5-0.5B-Instruct on two datasets - LightBlue and BGE-M3 - and compared BiXSE against two widely used pairwise losses:
> > > >
> > > > - Pairwise BCE, analogous to RankNet and PairDistill
> > > > - LambdaLoss, using NDCG-based gain weighting (v1 and v2)
> > > >
> > > > **Table 1: NDCG@10 (LightBlue training, in-batch only, batch size = 256)**
> > > >
> > > > | Loss Function           | BEIR (short) | BEIR  | MTEB (Multilingual v1) | TREC (qrel 19–23) | TREC 2021 (top-100) | TREC 2022 (top-100) | TREC 2023 (top-100) |
> > > > |------------------------|--------------|-------|--------------------------|-------------------|---------------------|---------------------|---------------------|
> > > > | **BiXSE (ours)**        |  73.91     |  45.05  | 55.46                 |  57.89          |  66.55            | 36.85              | 38.04              |
> > > > | Pairwise BCE           | 72.17        | 43.30 | 53.50                   | 55.74            | 65.02              | 37.58              | 37.29              |
> > > > | LambdaLoss (NDCG v1)   | 72.43        | 41.40 | 53.46                   | 55.25            | 65.04              | 37.68              | 38.72              |
> > > > | LambdaLoss (NDCG v2)   | 73.78        | 44.68 | 55.62                   | 56.39            | 64.84              | 37.25              | 38.23              |
> > > >
> > > > **Table 2: MTEB (English v2) NDCG@10 vs. batch size / hard negative count (BGE-M3 training)**
> > > >
> > > > | # Hard Negs / Batch Size | BiXSE (BCE) | Pairwise BCE | LambdaLoss (NDCG v2) |
> > > > |--------------------------|-------------|---------------|------------------------|
> > > > | 15 / 16                  | 44.09       | 40.89         | 48.42                  |
> > > > | 7 / 32                   | 45.45       | 45.62         | 49.92                  |
> > > > | 3 / 64                   | 46.60       | 44.05         | 51.91                  |
> > > > | 1 / 128                  | 46.50       | 49.38         | 49.49                  |
> > > > | 0 / 256 (in-batch only)  | 50.62       | 46.46         | 47.73                  |
> > > >
> > > > **Table 3: BEIR NDCG@10 vs. batch size / hard negative count (BGE-M3 training)**
> > > >
> > > > | # Hard Negs / Batch Size | BiXSE (BCE) | Pairwise BCE | LambdaLoss (NDCG v2) |
> > > > |--------------------------|-------------|---------------|------------------------|
> > > > | 15 / 16                  | 43.63       | 41.68         | 49.84                  |
> > > > | 7 / 32                   | 44.95       | 44.04         | 50.55                  |
> > > > | 3 / 64                   | 46.29       | 45.82         | 51.82                  |
> > > > | 1 / 128                  | 45.92       | 51.06         | 50.84                  |
> > > > | 0 / 256 (in-batch only)  | 48.88       | 45.66         | 45.49                  |
> > > >
> > > > These results demonstrate that BiXSE remains competitive or superior across multiple supervision densities. Importantly, while pairwise methods often require multiple labeled negatives per query to perform well, BiXSE achieves strong performance with only a single graded pair per query, made possible by in-batch negatives and the smooth supervision offered by BCE.
> > > >
> > > > We believe these updates clarify our goals and provide a more thorough and honest evaluation of BiXSE relative to pairwise alternatives. The new results will be included in the Appendix of the camera-ready version, and we are grateful to the reviewer for motivating these deeper comparisons.

---

> > > > > ### Author Response · Authors · 2025-06-07
> > > > >
> > > > > ## Side-by-side proposed changes to Introduction
> > > > >
> > > > > **Paragraph 1**: Preserved to maintain the initial motivation of the paper, and problems with binary relevance (general positioning). **Unchanged.**
> > > > >
> > > > > **Paragraph 2**: Introducing graded relevance from the IR community and what it is (general positioning). **Unchanged.**
> > > > >
> > > > > **Paragraph 3**: Where to search for scalable graded relevance signals for training. From cross-encoders to zero-shot LLM rankers. **Added clarification at the beginning**: Earlier ranking supervision was done via trained cross-encoders.
> > > > >
> > > > > « Early efforts to provide with scalable graded relevance signals involved training cross-encoding rankers on annotated data, followed by using their judgments as pseudo-labels to be distilled into bi-encoders~\citep{hofstatter2021improvingefficientneuralranking,santhanam2022colbertv2,chen2024bgem3embeddingmultilingualmultifunctionality,huang-chen-2024-pairdistill}. More recently, prompting LLMs like GPT-4 to serve as zero-shot rankers has yielded surprisingly strong results,… »
> > > > >
> > > > > **Paragraph 4**: Explicitlty acknowledge pairwise and listwise methods to train on graded relevance. Introduce BiXSE as a competitive scalable pointwise alternative. Conceptual comparison.
> > > > >
> > > > >  « Prior to the emergence of LLM-based relevance scoring, graded supervision was primarily leveraged via training with listwise or pairwise objectives~\citep{ranknet,lambdaloss,qu-etal-2021-rocketqa,pmlr-v130-reddi21a,huang-chen-2024-pairdistill}. However, such methods often rely on labeling multiple hard negative documents per query, significantly increasing annotation cost when using powerful LLMs as teachers, which hinders scalability. The opportunity of LLM-generated large-scale graded relevance data, however, calls for revisiting training objectives to align with the increased costs of quality graded relevance data. In this work, we propose \textbf{Binary Cross-Entropy Sentence Embeddings} (BiXSE), a simple pointwise training method that directly optimizes a binary cross-entropy (BCE) loss on graded relevance scores. BiXSE interprets graded relevance scores as probabilities within the range $[0, 1]$ to represent relevance continuity from completely irrelevant (0) to absolutely relevant (1). Unlike pairwise or listwise objectives, which rely on multiple supervised comparisons per query, BiXSE scales efficiently by enabling competitive performance by just using a single labeled query-document pair per query, while capturing structure implicitly via in-batch negatives. »
> > > > >
> > > > > **Paragraph 5**: Revise contributions to include experiments against pairwise baselines.
> > > > >
> > > > >  « We validate BiXSE through extensive experiments across retrieval and sentence embedding benchmarks, demonstrating consistent gains over standard InfoNCE objectives. Furthermore, BiXSE is, to the best of our knowledge, the first pointwise training method to consistently match or outperform strong pairwise ranking baseline losses when training on LLM-labeled graded relevance datasets. BiXSE scales across architectures and languages, approaches the performance of much larger zero-shot LLM-based rankers, and exhibits improved robustness to label noise compared to InfoNCE. Notably, it benefits from learning across a wider spectrum of graded relevance and achieves peak performance even without aggressive data filtering, making it a strong and efficient alternative for training dense models on LLM-generated supervision. As graded relevance becomes increasingly easy to generate, we argue that BiXSE offers a practical, robust, and scalable training paradigm for the next generation of dense retrieval systems. »

---

> > > > > > ### Author Response · Authors · 2025-06-07
> > > > > >
> > > > > > ## Proposed Revised Abstract
> > > > > >
> > > > > > Neural sentence embedding models for dense retrieval typically rely on binary relevance labels, treating query-document pairs as either relevant or irrelevant. However, real-world relevance often exists on a continuum, and recent advances in large language models (LLMs) have made it feasible to scale the generation of fine-grained graded relevance labels. In this work, we propose \textbf{BiXSE}, a simple and effective pointwise training method that optimizes binary cross-entropy (BCE) over LLM-generated graded relevance scores. BiXSE interprets these scores as probabilistic targets, enabling granular supervision from a single labeled query-document pair per query. Unlike pairwise or listwise losses that require multiple annotated comparisons per query, BiXSE achieves strong performance with reduced annotation and compute costs by leveraging in-batch negatives. Extensive experiments across sentence embedding (MMTEB) and retrieval benchmarks (BEIR, TREC-DL) show that BiXSE consistently outperforms softmax-based contrastive learning (InfoNCE), and—remarkably—matches or exceeds strong pairwise ranking baselines when trained on LLM-supervised data. BiXSE offers a robust, scalable alternative for training dense retrieval models as graded relevance supervision becomes increasingly accessible.

---

> > > > > > > ### Author Response · Authors · 2025-06-10
> > > > > > >
> > > > > > > As the discussion period comes to a close, we would like to sincerely thank the reviewer for their constructive feedback and thoughtful engagement. We hope that the clarifications and additional experiments we’ve provided address the concerns raised. If the revisions are in line with the reviewer’s expectations, we would greatly appreciate any reconsideration of the evaluation.

---

### Official Review · Reviewer_Dk2A · 2025-05-15

**Rating:** 7
**Confidence:** 4
**Ethics Flag:** 1

**Summary:**

This paper introduces BiXSE, a new training objective for dense retrieval and sentence embedding models that leverages graded relevance labels distilled from large language models. Rather than relying on softmax-based contrastive losses like InfoNCE, BiXSE uses binary cross-entropy to predict relevance probabilities between query-document pairs. This approach supports both binary and graded supervision and is shown to be more robust to label noise, scalable across model sizes and languages, and more data-efficient by avoiding over-filtering. Empirical results across TREC-DL, BEIR, and MTEB benchmarks show consistent performance gains over InfoNCE and other recent training objectives.

**Questions To Authors:**

- Can BiXSE be extended to support ranking-based losses (e.g., listwise or pairwise) that also leverage graded labels while retaining robustness?
- Have you tested how LLM scoring variability (e.g., using GPT-4 vs. PaLM) affects training stability or generalization under BiXSE?
- Would it be possible to use BiXSE with only ordinal labels (e.g., 0–3) without normalizing to [0, 1] probabilities? Would that be more stable?
- Could you elaborate more on how task-conditioned batching interacts with graded relevance—are there edge cases where this batching hurts?
- Given the observed robustness of BiXSE to label noise, have you considered its potential as a student objective in semi-supervised settings with pseudo-labels?
- In settings where graded labels are not available, does BiXSE degrade gracefully to binary data? How would you recommend adapting it to low-resource scenarios?

**Reasons To Accept:**

BiXSE offers a practical and theoretically grounded solution for learning from LLM-generated graded supervision, addressing limitations of binary labeling in retrieval. It supports a backward-compatible and scalable training paradigm with minimal changes to model architecture or training infrastructure. The BCE formulation is shown to be robust to label noise, less dependent on large batch sizes or heavy data filtering, and more effective at distilling signals from high-resource LLM rankers. Extensive experiments across benchmarks and model backbones show clear and consistent improvements over InfoNCE and alternative methods, validating the core hypothesis that graded relevance supervision can be leveraged more effectively via a pointwise BCE loss.

**Reasons To Reject:**

Despite strong empirical results, the method still relies on graded relevance annotations, which require access to powerful LLM rankers or curated datasets. It remains unclear how well BiXSE generalizes when LLM-generated labels are inconsistent or noisy across domains. The method’s reliance on a logit bias term and tuning its learning rate introduces another layer of optimization complexity. While the approach is positioned as a general-purpose alternative to contrastive learning, experiments are focused only on IR and sentence embedding tasks; the extent to which it transfers to tasks like multi-hop QA or dialog retrieval is unknown. Finally, some theoretical intuitions about robustness and BCE gradient behavior are described but not formally analyzed.

---

> ### Author Response · Authors · 2025-06-03
>
> We appreciate Reviewer Dk2A’s detailed and constructive feedback, along with their insightful questions. Below, we address each point individually:
>
> 1. Reliance on graded relevance annotations: It is now common practice for state-of-the-art dense retrievers to leverage scores from powerful LLMs (e.g., Gemini Embeddings). Moreover, our method explicitly demonstrates robustness to noisy labels, as evidenced by the reviewer’s summary and Figure 2 (right) in our manuscript.
>
> 2. Optimization complexity with logit bias term: Appendix Figure 5 (right) illustrates the response of the logit bias learning rate hyperparameter, clearly displaying a manageable and intuitive reverse U-curve response, making it straightforward to tune in practice.
>
> 3. Generalization beyond IR and sentence embedding tasks: Multi-hop QA is indeed covered in our benchmarks through HotpotQA included in BEIR and MTEB(eng, v2). Additionally, dialogue retrieval is represented by StatcanDialogueDatasetRetrieval in MTEB(multi, v1), demonstrating broader applicability. We will add the analytic breakdown of benchmarks in the Appendix of the manuscript.
>
> 4. Formal theoretical analysis: We thank the reviewer for highlighting this. While formalizing the theoretical intuitions around robustness and BCE gradient behavior is beyond this paper’s scope, we believe this is a valuable avenue for future exploration. Would the reviewer mind discussing what kind of result they expected in a case such as ours?
>
> Responses to specific questions:
>
> - Ranking-based losses: Extending BiXSE to pairwise or listwise BCE losses with integrated trainable logit biases to mitigate label imbalance is an interesting future research direction.
>
> - LLM scoring variability: Investigating variability across LLM scorers (e.g., GPT-4 vs. PaLM) is outside the current scope but could offer valuable insights into robustness across different labeling strategies.
>
> - Ordinal labels without normalization: BiXSE inherently requires [0, 1] probabilities due to its binary cross-entropy formulation. Ordinal labels can always be converted into this range without information loss, facilitating standardized training across datasets with different ordinal scales, as demonstrated in our experiments (E5 with binary labels and LightBlue with graded labels).
>
> - Task-conditioned batching interactions: Task-conditioned batching does not negatively interact with graded relevance. Previous works, such as GritLM (1) and NVEmbed (2), have established clear benefits from this batching strategy, and we observed no detrimental edge cases in our studies.
>
> - Semi-supervised scenarios with pseudo-labels: This is an insightful suggestion and we agree that exploring BiXSE as a student objective in semi-supervised settings represents a promising future direction.
>
> - Adapting BiXSE to binary-only or low-resource scenarios: BiXSE gracefully degrades to binary-labeled scenarios, as validated extensively in our binary-labeled experiments. In low-resource scenarios, additional supervision from smaller models (e.g., ColBERT (3)) could further enhance training effectiveness.
>
> We will include clarifications and further discuss these points in the revised manuscript, and we thank Reviewer Dk2A again for their thoughtful review and valuable insights.
>
> (1)  Generative Representational Instruction Tuning (Muenninghof et al. 2024)
> (2) NV-Embed: Improved Techniques for Training LLMs as Generalist Embedding Models (Lee et al. 2024)
> (3) ColBERT: Efficient and Effective Passage Search via Contextualized Late Interaction over BERT (Khattab et al. 2020)

---

> > ### Comment · Reviewer_Dk2A · 2025-06-05
> >
> > Thank you for your response. I will maintain my rating.

---

### Official Review · Reviewer_t3xV · 2025-05-26

**Rating:** 7
**Confidence:** 4
**Ethics Flag:** 1

**Summary:**

The authors identify the issues of requiring hard binary decisions for many tasks during LLM learning.
They introduce a method of training encoders using graded relevance scores from humans. This approach aligns training more closely with human preferences compared to the hard binary decisions typically used.
Authors validate their results on a huge data set of TREC relevance judgements.
The proposed methods significantly out performed the baselines.

**Reasons To Accept:**

The collection of graded relevance scores from humans collected from the TREC-DL is a probably the largest possible collection available.
Author results clearly demonstrate the claims strengths.

The paper is well-written and easy to follow. The motivation, methodology, and results are clearly presented.

**Reasons To Reject:**

The authors did not discuss any of the timing results of their experiments, such as training time or inference time, which are crucial for understanding the computational efficiency of their approach.
Presumably, there is a significant computational penalty.
More quantitative analysis of computational overhead (e.g., latency, memory usage, FLOPs, GPU utilization) would be helpful, especially for large-scale retrieval scenarios.

The authors should discuss trade-offs involved in computational encoding.
Include more detailed analysis of computational costs, especially for large-scale deployments with and without batches.

---

> ### Author Response · Authors · 2025-06-03
>
> We thank Reviewer t3xV for their thoughtful review and recognition of our paper’s strengths. We particularly appreciate the constructive suggestion to include more detailed computational analyses, and we are glad to clarify these points.
> Regarding the computational overhead:
>
> 1. **Inference Time**: Our proposed method, BiXSE, leverages dense retrievers whose inference-time computational cost is identical to models trained with standard softmax-based (InfoNCE) objectives. Once trained, BiXSE dense encoders have no additional runtime overhead, enabling identical inference latency. On an NVIDIA H100 GPU, inference latency for ranking query-document pairs is approximately 17ms per pair in a batched setting and maximum context length 8192 for the Qwen2.5-0.5B-Instruct model (identical to InfoNCE).
>
> 2. **Training Time and Memory Usage**: During training, BiXSE uses binary cross-entropy (BCE) across batches of query-document pairs, similar to InfoNCE training. Empirically, we observe negligible differences in training speed between BiXSE and InfoNCE objectives. Specifically, the difference in training time per epoch between BiXSE and InfoNCE is under 0.5% under identical batch sizes and GPU configurations. GPU memory and utilization were practically identical.
>
> 3. **FLOPs and GPU Utilization**: In general, computational FLOPs per forward pass remain constant regardless of the training objective, since BiXSE only changes the loss function computations that occur after embedding extraction. Thus, FLOPs and GPU utilization are effectively unchanged, as the loss computation has the same complexity.
>
> We agree with the reviewer that computational analyses are critical, especially for large-scale deployments. We will gladly incorporate this detailed analysis and clearly discuss these computational aspects in the revised manuscript. We thank the reviewer again for highlighting this important aspect.

---

> > ### Comment · Reviewer_t3xV · 2025-06-05
> >
> > Thank for for addressing the comments.

---

### Decision · Program_Chairs · 2025-07-08

**Decision:**

Accept

**Comment:**

**Summary**

This paper addresses a key limitation in training dense retrieval and embedding models which over rely on binary relevance labels. The authors propose BiXSE to effectively incorporate graded relevance scores, potentially derived from human annotations or LLMs. This approach allows the model to better capture fine-grained preference signals and demonstrates improved robustness to label noise, data efficiency, and scalability across model sizes and languages.

**Clarity**

The paper is clearly written and well-organized. The authors support their claims with thorough experimental validation across a range of benchmarks, including TREC-DL, BEIR, and MTEB. The inclusion of new baselines during the discussion period (e.g., LambdaLoss) further strengthens the paper.

**Novelty**

While the idea of using cross-entropy loss for ranking is not novel (e.g., RankNet), and graded supervision has been explored in previous work (e.g., RankDistil) mentioned by some reviewers, the paper presents a practical and well-integrated formulation that adapts these ideas to the context of modern dense retrieval models and LLM-generated supervision. The proposed method offers a distinct contribution in its simplicity, generality, and empirical gains, even if it builds on familiar components.

**Pros**
- A practical training objective that supports both binary and graded supervision.
- Demonstrated improvements across multiple datasets and benchmarks.
- Clear exposition and strong empirical validation.
- Addresses an important pain point in retrieval model training (binary vs. graded relevance).

**Cons**
- The novelty is moderate; key ideas have some precedent in prior work (e.g., RankNet, RankDistil).
- Some initially missing baselines (e.g., pairwise methods) were only added later in the rebuttal.
- The claim of greater efficiency over prior distillation approaches could be more thoroughly analyzed.

**Recommendation**

All reviewers rated the paper between 6 and 7 (6, 7, 7, 7) and reached a consensus that the paper is solid and worthy of acceptance.
Despite concerns regarding novelty, the practical utility, empirical strength, and methodological clarity of the proposed approach justify inclusion in the conference.